# Novel viral splicing events and open reading frames revealed by long-read direct RNA sequencing of adenovirus transcripts

Alexander M. Price[1], Robert T. Steinbock[2], Richard Lauman[1,3], Matthew Charman[1], Katharina E. Hayer[4], Namrata Kumar[1], Edwin Halko[1], Krystal K. Lum[1], Monica Wei[2], Angus C. Wilson[5], Benjamin A. Garcia[6¤], Daniel P. Depledge[5,7,8], Matthew D. Weitzman[1,9]*

1 Division of Protective Immunity, Department of Pathology and Laboratory Medicine, The Children's Hospital of Philadelphia, Philadelphia, Pennsylvania, United States of America, 2 Cell & Molecular Biology Graduate Group, University of Pennsylvania, Philadelphia, Pennsylvania, United States of America, 3 Graduate Group in Biochemistry and Biophysics, University of Pennsylvania, Philadelphia, Pennsylvania, United States of America, 4 Department of Biomedical and Health Informatics, The Children's Hospital of Philadelphia, Philadelphia, Pennsylvania, United States of America, 5 Department of Microbiology, New York University School of Medicine, New York city, New York, United States of America, 6 Department of Biochemistry and Biophysics, University of Pennsylvania Perelman School of Medicine, Philadelphia, Pennsylvania, United States of America, 7 Institute of Virology, Hannover Medical School, Hannover, Germany, 8 German Center for Infection Research (DZIF), partner site Hannover-Braunschweig, Hannover, Germany, 9 Department of Pathology and Laboratory Medicine, University of Pennsylvania Perelman School of Medicine, Philadelphia, Pennsylvania, United States of America

¤ Current address: Department of Biochemistry and Molecular Biophysics, Washington University School of Medicine, St. Louis, Missouri, United States of America
* weitzmanm@chop.edu

**Data Availability Statement:** Basecalled fast5 (Nanopore) and fastq (Illumina) datasets generated as part of this study can be downloaded from the

## Abstract

Adenovirus is a common human pathogen that relies on host cell processes for transcription and processing of viral RNA and protein production. Although adenoviral promoters, splice junctions, and polyadenylation sites have been characterized using low-throughput bio-chemical techniques or short read cDNA-based sequencing, these technologies do not fully capture the complexity of the adenoviral transcriptome. By combining Illumina short-read and nanopore long-read direct RNA sequencing approaches, we mapped transcription start sites and RNA cleavage and polyadenylation sites across the adenovirus genome. In addition to confirming the known canonical viral early and late RNA cassettes, our analysis of splice junctions within long RNA reads revealed an additional 35 novel viral transcripts that meet stringent criteria for expression. These RNAs include fourteen new splice junctions which lead to expression of canonical open reading frames (ORFs), six novel ORF-containing transcripts, and 15 transcripts encoding for messages that could alter protein functions through truncation or fusion of canonical ORFs. In addition, we detect RNAs that bypass canonical cleavage sites and generate potential chimeric proteins by linking distinct gene transcription units. Among these chimeric proteins we detected an evolutionarily conserved protein containing the N-terminus of E4orf6 fused to the downstream DBP/E2A ORF. Loss of this novel protein, E4orf6/DBP, was associated with aberrant viral replication center morphology and poor viral spread. Our work highlights how long-read sequencing technologies

European Nucleotide Archive (ENA) under the following study accessions: PRJEB35667 and PRJEB35652. Raw MS files associated with this work have been deposited to the public database the ProteomeXchange Consortium repository via the PRIDE partner repository. The accession number for the MS data is PXD025339 and PXD034464. The newly generated genome and transcriptome annotation can be found at https://github.com/DepledgeLab/Ad5-annotation and under the GenBank accession number OP218818. The authors declare that all other data supporting the findings of this study are available within the manuscript and its Supplementary Information files.

**Funding:** This work was supported through NIH grants R21-AI130618 and R21-AI147163 (ACW), and R21-AI54654, R01-AI145266, and R01-AI121321 (MDW), and R01-AI118891 (MDW and BAG). Additional support came from the NCI T32 Training Grant in Tumor Virology T32-CA115299 (AMP), Individual National Research Service Award F32-AI138432 (AMP), and Pathway to Independence award K99-AI159049 (AMP). The funders had no role in study design, data collection and analysis, decision to publish, or preparation of the manuscript.

**Competing interests:** The authors have declared that no competing interests exist.

combined with mass spectrometry can reveal further complexity within viral transcriptomes and resulting proteomes.

## Author summary

Adenoviruses are important human pathogens that are also used as powerful tools for gene delivery and as vaccine agents. In addition, studies of the molecular biology of adenovirus have led to seminal discoveries in cell biology, RNA biology, and pathogenesis. While adenoviruses, have been exceptionally well-characterized over the past decades of research, much is still left to discover about their biology. By applying long-read RNA sequencing we have comprehensively reannotated the transcriptome of the Ad5 serotype. In doing so, we discovered an additional 35 abundant viral transcripts that potentially encode for 20 novel viral open reading frames. We demonstrate that one of these novel proteins joins two disparate genetic units into a chimeric protein that is evolutionarily conserved among adenoviruses. Discovery of this protein, E4orf6/DBP, recontextualizes prior studies on the function of the critical viral effector E4orf6. Furthermore, E4orf6/DBP is shown to modulate the morphology of membrane-less viral replication centers in the nucleus and function in cell-to-cell spread of the virus. Together, we showcase how modern techniques in RNA sequencing and mass spectrometry can reveal additional complexity in a well-studied model viral pathogen.

## Introduction

Adenoviruses (AdV) are common viral pathogens across multiple species with distinct tissue tropisms including gut, eye, and lung [1]. Among the human adenoviruses, serotypes 2 (Ad2) and 5 (Ad5) from subgroup C are the most prevalent, and cause benign to severe respiratory infections [2]. These two serotypes are highly homologous, sharing 94.7% nucleotide homology between their genomes and 69.2–100% amino acid identity amongst conserved open reading frames (ORFs) [3,4]. AdVs readily infect most transformed human cell lines, and for many decades have served as valuable tools contributing to many seminal discoveries in molecular biology [5]. RNA splicing was discovered by the analysis of adenovirus encoded RNAs [6,7], as well as other important findings in messenger RNA capping and polyadenylation [8,9]. It is now understood that essentially all AdV mRNAs are capped, spliced, polyadenylated, and exported from the nucleus using host cell machinery [10].

AdV are capable of infecting non-dividing cells and reprogramming cellular processes for productive viral infection. This rewiring involves a highly regulated cascade of viral gene expression over various kinetic classes [5]. The first viral gene to be expressed after infection is E1A, a multi-functional transcription factor that activates downstream viral transcription, liberates E2F from RB proteins, and alters host transcriptional responses to the virus [11–14]. While all E1A molecules have identical 5' and 3' nucleotide sequences, splicing of differently sized internal introns allows for the production of discrete proteins that lack specific functional domains conserved across serotypes [15]. Early after infection, E1A is expressed mainly as large and small isoforms, but later in infection alternative splicing leads to the production of a 9 Svedberg E1A isoform (E1A-9s) as well as low abundance doubly-spliced E1A-11s and E1A-10s. The second viral gene to be activated after infection is E1B, predominantly consisting of two spliced isoforms producing 19-kilodalton and 55-kilodalton proteins, with two less

abundant isoforms potentially encoding polypeptides of 156 and 93 residues [16]. While E1B-19K acts to block cellular apoptosis [17], E1B-55K is another multifunctional protein that can cooperate with E1A to alter cellular gene expression downstream of p53, as well as form the targeting component of a viral ubiquitin ligase [18–23]. The remaining early transcription units are all transcriptionally activated by E1A and encode products of related function. The E2 region on the reverse strand of the AdV genome has both an early and a late promoter, as well as two distinct polyadenylation sites, leading to upstream E2A and downstream E2B transcripts [24]. E2A encodes for the viral DNA-binding protein (DBP), while alternative splicing to E2B encodes for the protein-priming terminal protein (pTP), as well as the AdV DNA polymerase (AdPol) [25–27]. The E3 region encoded on the top strand also has two polyadenylation sites leading to E3A and E3B transcription units. The E3 gene products are primarily involved in modulating the host innate immune system [28–30]. Like E1A, the E4 region on the reverse strand has identical 5' and 3' regions, and encodes up to six different ORFs by removal of a first intron of varying length. E4 region transcripts encode for multifunctional proteins that are involved in the regulation of transcription, splicing, and translation of viral RNAs, as well as antagonizing intrinsic cellular defenses [31–33]. Additionally, AdV encodes two Pol III-derived virus associated (VA) RNAs involved in the inactivation of Protein Kinase acting on RNA (PKR) [34,35]. Ultimately, the concerted efforts of the AdV early proteins lead to a cellular state that allows for the replication and amplification of the viral DNA genome [36,37].

Prior to viral DNA replication, the AdV Major Late Promoter (MLP) is thought to be largely silent with small amounts of RNA being made that terminate at the first downstream (L1) polyadenylation site [38]. At this time, so-called intermediate genes pIX and IVa2 begin to be expressed from promoters within the E1B cassette and antisense to the MLP. Both pIX and IVa2 co-terminate at polyadenylation sites within the early genes that they overlap with (E1B and E2B, respectively) and are involved in late gene transcription and packaging [39,40]. Only after viral DNA replication has occurred does the MLP fully activate, supporting the hypothesis that active replication *in cis* is a prerequisite for full viral late gene expression [41–43]. The Major Late Transcriptional Unit (MLTU) begins with a series of three constitutive exons spliced together to form the tripartite leader, before downstream splicing to late cassettes defined by one of five alternative polyadenylation sites (termed L1-L5) [38]. Splicing within the tripartite leader to the so-called "i" exon leads to a putative ORF upstream of subsequent late gene splicing events and destabilizes these RNA molecules [44,45]. An additional intermediate promoter has been reported within the L4 region that allows for the early expression of L4-22K and L4-33K proteins important for the splicing of other late genes [46,47]. The MLTU encodes for primarily structural capsid components or proteins involved with packaging of new virions, and their expression ultimately leads to the death of the host cell. More recently, a novel late gene, UXP, was discovered on the reverse strand of the genome [48,49]. The UXP promoter is located between E4 and E2, and splices downstream to the exons within the E2A region to continue translation of an ORF in an alternate reading frame to that of DBP. This exciting finding suggests that our knowledge of AdV transcripts is incomplete, especially within the complex MLTU region.

The complete Ad5 genome was sequenced in 1991 using Sanger sequencing of viral genome fragments inserted into plasmid DNA and amplified in bacteria [3]. This genome sequence was then annotated in 2003 based on homology to similar serotypes of AdV [4]. As such, the current reference annotation for Ad5 available on the National Center for Biotechnology Information (AC_000008) is incomplete, and lacks critical information such as transcription start sites (TSS), cleavage and polyadenylation sites (CPAS), and the resulting 5' and 3' untranslated regions (UTR) that the aforementioned information dictates. In recent years,

new technologies have allowed for high-throughput investigation of gene expression utilizing various techniques. The effect of AdV infection on host gene expression has been shown for Ad5 by microarray analysis [50,51], as well as for Ad2 by Illumina-based short-read sequencing [52,53]. Analyses of both single-end and paired-end short-read RNA-seq data from cells infected with Ad2 revealed both temporal viral gene expression and high-depth splicing information, and identified both previously confirmed and novel RNA splice site junctions [54]. In addition, temporal analysis of Ad5 viral gene expression was performed using digital PCR to determine expression kinetics of a subset of known viral genes [55]. The late RNA tripartite leader splicing was also analyzed by short-read sequencing across a number of human AdV serotypes [44].

While the quality and depth of current short-read sequencing technologies are high, the complex nature of many viral transcriptomes precludes the unambiguous mapping of these short reads to any one particular RNA isoform due to extreme gene density and overlapping transcriptional units [56,57]. In this regard, the ability of long-read RNA sequencing to map full-length transcripts has the potential to revolutionize detection of novel isoforms and multiply spliced RNA at the single-molecule level [58–60]. Two recent studies have leveraged the power of Oxford Nanopore Technologies to characterize an exceptional diversity of adenoviral splicing [61,62]. Donovan-Banfield et al., performed direct RNA sequencing over a time course of infection with Ad5 in MRC5 fibroblast cells and characterized over 11,000 different splicing patterns [61]. Their study used an "ORF-centric" approach to define low level splice sites leading to altered production of canonical genes, as well as multiple upstream and alternative open reading frames. In contrast, Westergren-Jakobssen et al., used direct cDNA based sequencing on a time course of Ad2-infected IMR90 lung fibroblast cells [62]. This study uncovered over 900 potential RNA transcripts, including upstream transcription from the inverted terminal repeats, and compared novel ORFs to existing Ad2 proteomics data. While the diversity and complexity of potential adenovirus splicing is immense, it remains difficult to determine functional relevance of lowly expressed alternative transcript isoforms.

In this study, we have re-annotated the Ad5 genome and transcriptome using a combination of short-read and long-read RNA sequencing technologies. The high read depth and accuracy of base-calling achieved by Illumina-based short-read sequencing allowed for both the detection of single nucleotide polymorphisms within transcriptionally active regions of the viral DNA genome, as well as error-correction of the inherently noisier base-calling of Nanopore-based long-read direct RNA sequencing (dRNA-seq). Detection of full-length RNA transcripts and the assignment of TSS and CPAS transcriptome-wide was enabled by dRNA-seq. Furthermore, by combining highly accurate splice site junctions from short-read sequencing and full-length isoform context from long-read sequencing, we were able to reevaluate the splicing complexity of AdV transcriptional units. Using this integrated approach, we have discovered 35 additional viral polyadenylated RNAs for a total of 90 unique mRNAs produced by Ad5. Of these novel isoforms, 14 RNAs encode for a canonical ORF with changes in upstream or downstream splicing. The remaining 21 encode new ORFs or alter existing ORFs by internal truncations or in-frame fusion of genes from separate transcription units. In particular, we have focused on one novel transcript, a fusion of the N-terminus of E4orf6 to the downstream DBP ORF which generates an abundant and evolutionarily conserved chimeric protein that we called E4orf6/DBP. We show that E4orf6/DBP regulates the morphology of viral replication centers and ultimately impacts viral spread. Taken together, our data reveal exceptional transcriptional complexity of AdV and highlight the necessity of revisiting transcriptome annotations following the emergence of appropriate new technologies.

## Results

### RNA-seq reveals high-confidence SNPs within the Ad5 genome

Illumina-based RNA sequencing (RNA-seq) relies on the fractionation of RNA molecules before reverse transcription into complementary DNA, and therefore loses information such as RNA modifications and the context of distal splice junctions within full-length molecules. However, the accuracy of each individual base call is very high [63]. Using bcftools, a common variant-calling algorithm designed to assess allele-specific variation within RNA-seq, we were able to detect single nucleotide polymorphisms (SNPs) within the RNA transcriptome that likely emerge from mutation within the DNA genome [64,65]. While RNA modifications such as inosine can be read as SNPs during the process of reverse transcription, these events did not approach the 99% read depth stringency we required among our three biological replicates to call a conserved variant [66]. While this technique will only capture SNPs within the actively transcribed region of the genome, nearly every nucleotide of the gene-dense AdV genome is transcribed at a sufficient level for this strategy to provide meaningful data.

In total, we discovered 24 SNPs and no insertions or deletions in the Ad5 genome when compared to the original annotation (Fig 1). Of these deviations, exactly half (12) are not predicted to change amino acid coding capacity, with two SNPs occurring within untranslated regions of viral RNA and the other ten leading to synonymous amino acid codons within all reading frames annotated to be protein producing. The remaining 12 mutations are predicted to lead to coding sequence variations at the amino acid level, with all examples being missense mutations and no evidence of premature stop codons. Importantly, none of the mutations discovered generated novel RNA splice sites. These data demonstrate the ability to call mutations within the DNA genomes of viruses using solely high-depth RNA sequencing data. While these SNPs could have arisen during cell culture passage solely within our lab, whole-plasmid sequencing of a common Ad5 mutagenesis platform (pTG3602, kind gift of Patrick Hearing) revealed exactly the same 24 SNPs with no additions or Indels. Furthermore, detecting only 24 SNPs out of 35,938 nucleotides highlights the overall genomic stability of AdV.

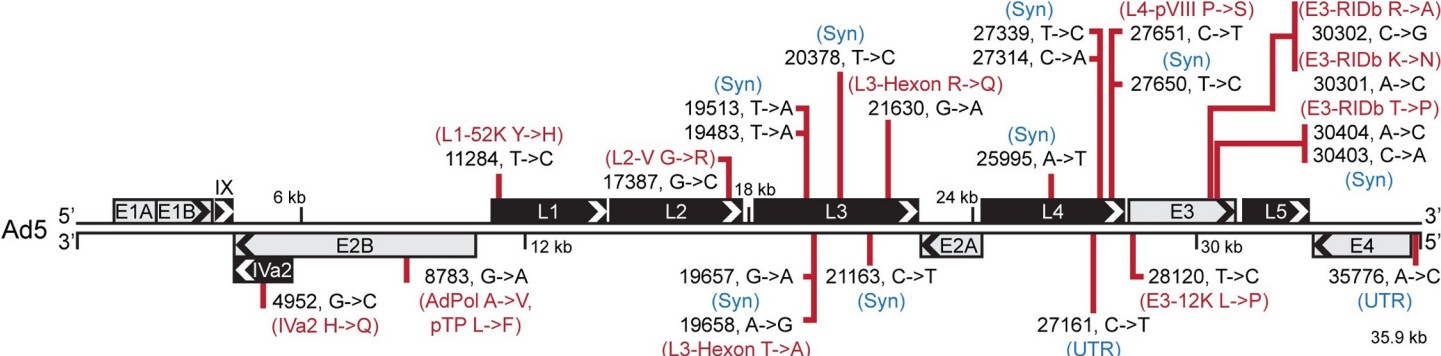

**Fig 1. RNA-seq reveals high-confidence SNPs within the Ad5 genome.** The 35,938 base pair linear genome of Ad5 is displayed in the traditional left to right format. Major transcriptional units are shown as boxes above or below the genome with arrowheads denoting the orientation of the open reading frames (ORFs) encoded within. Grey boxes denote early gene transcriptional units while black boxes denote late genes. Bcftools was used to analyze short-read RNA seq data to predict single nucleotide polymorphisms (SNPs) and insertions/deletions (InDels) that approach 100% of the RNA reads when compared to the reference Ad5 genome (AC_000008). In total, 24 such SNPs were discovered and their positions within the genome is highlighted by red vertical lines. For each SNP, the nucleotide position as well as the top strand reference base and corrected base are shown in black text (nucleotide position, reference base -> corrected base). If indicated SNPs fell within untranslated regions (UTR), or did not change the encoded amino acid of any annotated reading frame potentially impacted by the SNP, these were marked with blue text denoting either UTR or Syn (synonymous mutation), respectively. For any SNP that led to an amino acid change within an annotated ORF, these ORFs as well as the identity of the reference amino acid and corrected amino acid are highlighted in red.

## Combined short-read and long-read sequencing showcases adenovirus transcriptome complexity

To compare short-read Illumina sequencing and long-read nanopore sequencing directly, A549 cells were infected with Ad5 for 24 hours and total RNA was harvested in biological triplicate. Three samples were prepared into standard strand-specific Illumina RNA-seq libraries targeting the polyadenylated mRNA fraction. The same RNA samples were then pooled and poly(A) purified before submitting to direct RNA sequencing (dRNA-seq) on an Oxford Nanopore Technologies MinION MkIb platform [67]. Resulting sequence reads were aligned to the Ad5 reference genome using either GSNAP for short-reads [68], or MiniMap2 for long-reads [69]. Overall sequencing depth for both forward and reverse reads are shown in **Fig 2**. While Illumina sequencing provided on average three times the read depth when compared to dRNA sequencing, the overall coverage plots were similar.

dRNA-seq is performed in the 3' -> 5' direction and thus allows precise mapping of the 3' ends of transcripts at which poly(A) tails are added (cleavage and polyadenylation site, CPAS) [67]. Where the quality of input RNA is high, a variable proportion of sequence reads extend all the way to their transcription start site (TSS). By collapsing sequence reads to their 5' and 3' ends, we were able to implement a peak-calling approach to predict TSS and CPAS [58], and map their positions along both the forward and reverse strand of the viral genome (**Fig 2**). In addition, ContextMap2 [70] was used to mine Illumina RNA-seq data for short read sequences containing poly(A) stretches that could be aligned against the viral genome for an orthogonal method of CPAS detection (**Fig 2**). Mapping the TSS on the forward strand revealed the locations of the promoters for E1A, E1B, pIX, MLP, and E3, while the reverse strand revealed the promoters for E4, UXP, E2-early, E2-late, and IVa2. While we were able to reliably detect the L4 promoter TSS at 12 hpi, this was not clearly detected above noise at 24 hpi [46]. When mapping CPAS loci, we saw great concordance between the dRNA-seq and ContextMap2 performed on short-read sequences. On the forward strand we were able to detect previously mapped CPAS events at the E1A, E1B/pIX, E3A, E3B, and individual L1 through L5 sites. On the reverse strand we detected CPAS at the E4, UXP/DBP, and E2B/IVa2 locations. Exact nucleotide locations of both TSS and CPAS can be found in **S1 Table**. In addition, we also detected TSS and CPAS around the RNA pol III-derived VA RNA I (**Fig 2**). While pol III transcripts are generally not polyadenylated, and thus would not be captured by our nanopore sequencing approach, it was previously reported that low levels of polyadenylation can occur on these transcripts [71]. Given the high abundance of AdV VA RNAs (up to $10^8$ copies per cell during late infection), it remains likely that low level VA RNA polyadenylation events are occurring [35].

To generate accurate splicing maps of AdV transcripts, we combined the sensitivity of short-read sequencing to identify RNA junctions and then placed them in the context of full-length RNA isoforms using dRNA sequencing. Due to the spurious nature of low-level AdV splicing events [54], we set abundance thresholds for the highly abundant viral late transcripts of 500 reads for short-read junctions, and at least ten events detected in the long-read sequencing when collapsed by FLAIR [72]. Using this method, we readily detected other recently discovered viral isoforms, such as multiple splice sites preceding the pVII ORF [54], the so-called X, Y, and Z leaders embedded in E3 and preceding L5-Fiber [54,73], and the newly described UXP [48,49]. Using full-length RNAs, we were able to detect novel splice sites producing canonical ORFs that only differ in UTRs for E1A, L4-100K, L4-33K, L4-pVIII, DBP, and E4orf6/7. In addition, we discovered canonical ORF isoforms embedded within transcripts generated from non-canonical promoters, such as Fiber driven by the E3 promoter, multiple E3 proteins driven by the Major Late Promoter, and DBP driven by the E4 promoter. Within

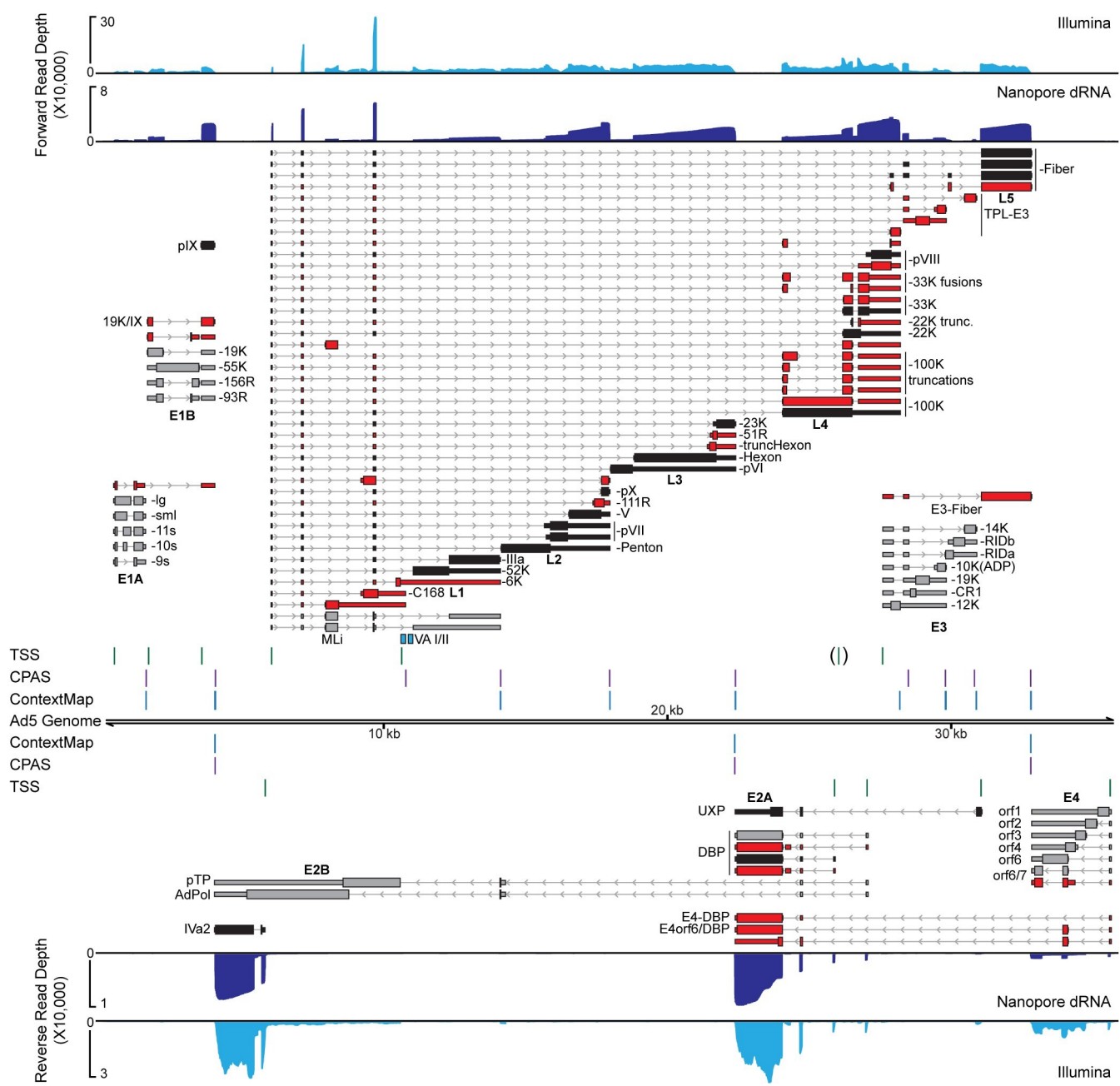

**Fig 2. Combined short-read and long-read sequencing showcases adenovirus transcriptome complexity.** A549 cells were infected with Ad5 for 24 hours before RNA was extracted and subjected to both short-read and long-read sequencing. Sequence coverage provided by short-read stranded RNA-seq (Illumina, light blue), as well as nanopore long-read direct RNA-seq (Nanopore dRNA, dark blue), is shown along the Ad5 genome. For both tracks, reads aligning to the forward strand are plotted above the genome, while reads aligning to the reverse strand are shown below the genome. For dRNA-seq datasets, reads can be reduced to their 5' and 3' ends and peak-calling applied to predict individual transcription start sites (TSS, green vertical lines) or cleavage and polyadenylation sites (CPAS, magenta vertical lines), respectively. The TSS for the L4 promoter was only detected at 12 hpi and is thus displayed in brackets. Similarly, the ContextMap algorithm can predict, albeit at lower sensitivity, CPAS sites from poly(A) containing fragments within Illumina RNA-seq data (ContextMap, light blue vertical lines). Individual RNA transcripts are shown above and below the genome, with thin bars denoting 5' and 3' untranslated regions (UTR), thick bars denoting open reading frames (ORFs), and thin lines with arrowheads denoting both introns and orientation of transcription. Previously characterized early genes are denoted in grey, while previously characterized late genes are denoted in black. RNA isoforms discovered in this study are highlighted in red. Names of transcriptional units are shown under each cluster of transcripts, while the name of the protein derived from the respective ORF is listed after each transcript to the right or left. The position of Pol III-derived noncoding RNAs virus associated VA-I and VA-II are highlighted in teal boxes.

transcriptional units, we discovered the presence of internal splice sites leading to in-frame truncations of existing ORFs, such as E1B-19K, L3-Hexon, L4-22K, and four distinct isoforms of truncated L4-100K. We also discovered splicing events predicted to lead to in-frame fusion events within transcriptional units, such as fusions between N-terminal fragments of L4-100K and L4-33K or L4-pVIII or the X-Z-Fiber ORF. Furthermore, gene fusion events were observed that join disparate transcriptional units, such as an N-terminal fragment of E1B-19K and pIX (19K/IX), the internal major late exon to L4-100K (MLi/100K), and E4orf6 to DBP (E4orf6/DBP). Lastly, we detected multiple transcripts that could encode novel ORFs with a minimum of 50 amino acids. Two ORFs, C168 and L1-6K, have been predicted in the related serotype Ad2 but there was only evidence for production of C168. Novel splice acceptors within L2 and L3 lead to RNAs containing 111 residue and 51 residue ORFs (L2-111R and L3-51R, respectively). Both of these splice sites are conserved in Ad2 [54,62]. Finally, a transcription unit crossing RNA joining the E4orf6/7 splice donor to the major two exons within E2A leads to a potential 133 residue ORF (E4-Unk) that would be translated in an alternate frame from both DBP and UXP. The complete splicing map of Ad5 is included as **S1 Fig**. Overall, we discovered 35 new isoforms for a total of 90 expressed RNA isoforms during Ad5 infection. When compared to recently published adenovirus long-read sequencing studies [61,62], we find concordance for 33/35 transcripts from the matching Ad5 strain and 28/35 transcripts from the related Ad2 strain (**S2 Table**).

## Detection of novel Ad5 protein species utilizing mass spectrometry

Our laboratory recently generated a high depth whole cell proteome (WCP) using fractionation and label-free quantification during a time-course of Ad5 infection [74]. In this study, canonical viral proteins were detected and quantified comprehensively, so we reasoned that our existing WCP contained enough depth to provide evidence for the existence of novel viral ORFs predicted by RNA sequence. While there are a total of 20 novel ORFs predicted by this study, 15 of them are fusions between otherwise canonical AdV ORFs. Therefore, for these 15 chimeric transcripts only a single unique tryptic peptide covering the junction region defines the unique protein from the shared peptides of other canonical AdV viral proteins. In addition, Parallel Reaction Monitoring Mass Spectrometry (PRM-MS) was used to detect specific peptides of known mass and charge to validate many of the lower abundance peptide junctions identified using the Data Dependent Acquisition-based fractionated dataset. In total, our mass spectrometry analysis found high confidence evidence for 12/20 novel ORFs (60%) and is summarized in **Table 1**. Thus, it is likely that many more of the novel AdV ORFs exist, but were simply not detected due to unfavorable mass and charge characteristics coupled with fusion proteins only possessing a single informative junctional peptide. While these novel proteins are uniformly low in abundance compared to the canonical viral proteins, they might affect the function of the canonical proteins they are composed of in dominant negative ways by swapping of key domains. In addition, these novel peptides might be loaded into MHC molecules for signaling to the adaptive immune system [75].

## Direct RNA Sequencing unambiguously distinguishes early and late transcription

We next determined if we could provide unambiguous detection of viral transcripts over a time-course of infection that captured early and late viral kinetics. By aligning long reads to the fully re-annotated viral transcriptome (as opposed to the viral genome), and only counting the reads that could be unambiguously assigned to a single transcript, we were able to detect all of the canonical and newly discovered transcripts (**Fig 3**). At 12 hours post-infection (hpi)

**Table 1. Unique peptides of novel AdV ORFs detected by mass spectrometry.** Specific detected viral peptides are listed along with their amino acid number in brackets. PSM, peptide spectral match within whole cell proteome. PRM Detection, parallel reaction monitoring run performed. PF/TF, peptide fragments/total possible peptide fragments detected by PRM.

| Protein Name | Peptide | PSM | q-Value | PRM Detection | PF/TF |
|---|---|---|---|---|---|
| E1B-19K/IX | [57–81] SCAAAAAMSTNSFDGSIVSSYLTTR | Nd | | Yes | 20/25 |
| MLi | [1–23] MRADREELDLPPPIGGVAIDVVK | 15 | 2.37E-05 | | |
| | [3–23] ADREELDLPPPIGGVAIDVVK | 187 | 2.37E-05 | | |
| | [3–31] ADREELDLPPPIGGVAIDVVKVEVPATGR | 24 | 2.37E-05 | | |
| | [6–23] EELDLPPPIGGVAIDVVK | 35 | 8.26E-05 | | |
| | [24–31] VEVPATGR | 74 | 0.000167147 | | |
| | [32–39] TLVLAFVK | 46 | 0.000139108 | | |
| | [40–63] TCAVLAAVHGLYILHEVDLTTAHK | 110 | 2.37E-05 | | |
| | [64–75] EAEWEFEPLAWR | 12 | 2.37E-05 | | |
| | [107–113] AQSPDVR | 47 | 0.000433765 | | |
| | [116–136] RSELDDNIAQMGAVHGLELPR | 71 | 2.37E-05 | | |
| | [116–137] RSELDDNIAQMGAVHGLELPRR | 3 | 2.37E-05 | | |
| | [117–136] SELDDNIAQMGAVHGLELPR | 53 | 2.37E-05 | | |
| C168 | [120–134] DLSESASTGSENLSR | 2 | 0.000783189 | | |
| L1-6K | [1–16] MYLDIQVMPAAVVEAR | 1 | 0.00836917 | | |
| L2-111R | [77–93] SLWFLQIWPSPAASVSR | 4 | 8.26E-05 | | |
| | [94–111] CRDSEEECTVGGAWPATA | 6 | 2.37E-05 | No | |
| L3-51R | [38–47] EASNINNSCR | 2 | 0.00530179 | | |
| L4-100Ktr3 | [60–81] SVPTEDKKQDQDNAEPYQQQPR | 8 | 2.37E-05 | | |
| | [67–81] KQDQDNAEPYQQQPR | 10 | 2.37E-05 | | |
| | [68–81] QDQDNAEPYQQQPR | 8 | 2.37E-05 | No | |
| L4-100Ktr4 | [155–172] LNFYPVFAVPEPYQQQPR | 20 | 2.37E-05 | Yes | 13/18 |
| MLi-100K | [137–145] RQPYQQQPR | 16 | 8.26E-05 | | |
| L4-100K-33K | [93–98] GPLLPR | 13 | 0.00496589 | Yes | 3/6 |
| E4-Unk | [94–106] SSAKPPPSADAVR | 3 | 0.00343938 | | |
| E4orf6/DBP | [39–60] ATILEDHPLLPECNTLTMHNEK | Nd | | Yes | 17/22 |

the majority of viral transcripts detected were early RNAs, particularly E1A-large and E1A-small, E1B-19K and E1B-55K, early promoter DBP, E3-12K, E3-19K, and E4orf3 (**Fig 3A**). However, at this time point we still detected low-level viral late transcripts that progressed beyond the L1 polyadenylation site, corroborating recent observations [55]. At 24 hpi, however, viral gene expression shifted to be dominated by late gene expression, as well as early transcripts derived from the E1B and late promoter DBP locus (**Fig 3B**). At late times post-infection we also saw the E3-Fiber transcript, as well as the tripartite leader containing E3 locus. The transcripts spanning disparate transcriptional units, i.e., 19K/IX (14-fold), E3-Fiber (15-fold), and E4orf6/DBP (6-fold), increase substantially at 24 hpi, implicating these messages as novel late transcripts, with expression as abundant as the recently described late UXP transcript [48,49]. Furthermore, while all permutations of the X, Y, and Z leaders preceding Fiber were previously detected by short-read sequencing, these could not be phased to full-length transcript isoforms [54,73]. Our full-length RNA data indicate that all Fiber transcripts can be detected, but MLP-Fiber and Y-Fiber are the most abundant, followed by XY-Fiber, and then all other isoforms. While the previous lack of detection of some of these novel transcripts can be explained by low overall abundance (e.g., L2-111R, L4-100K/VIII), many of the L4-100K truncations and L4-33K fusions are expressed at levels higher than that of the *bona fide* late transcript UXP. These data demonstrate that the newly discovered viral transcripts can be

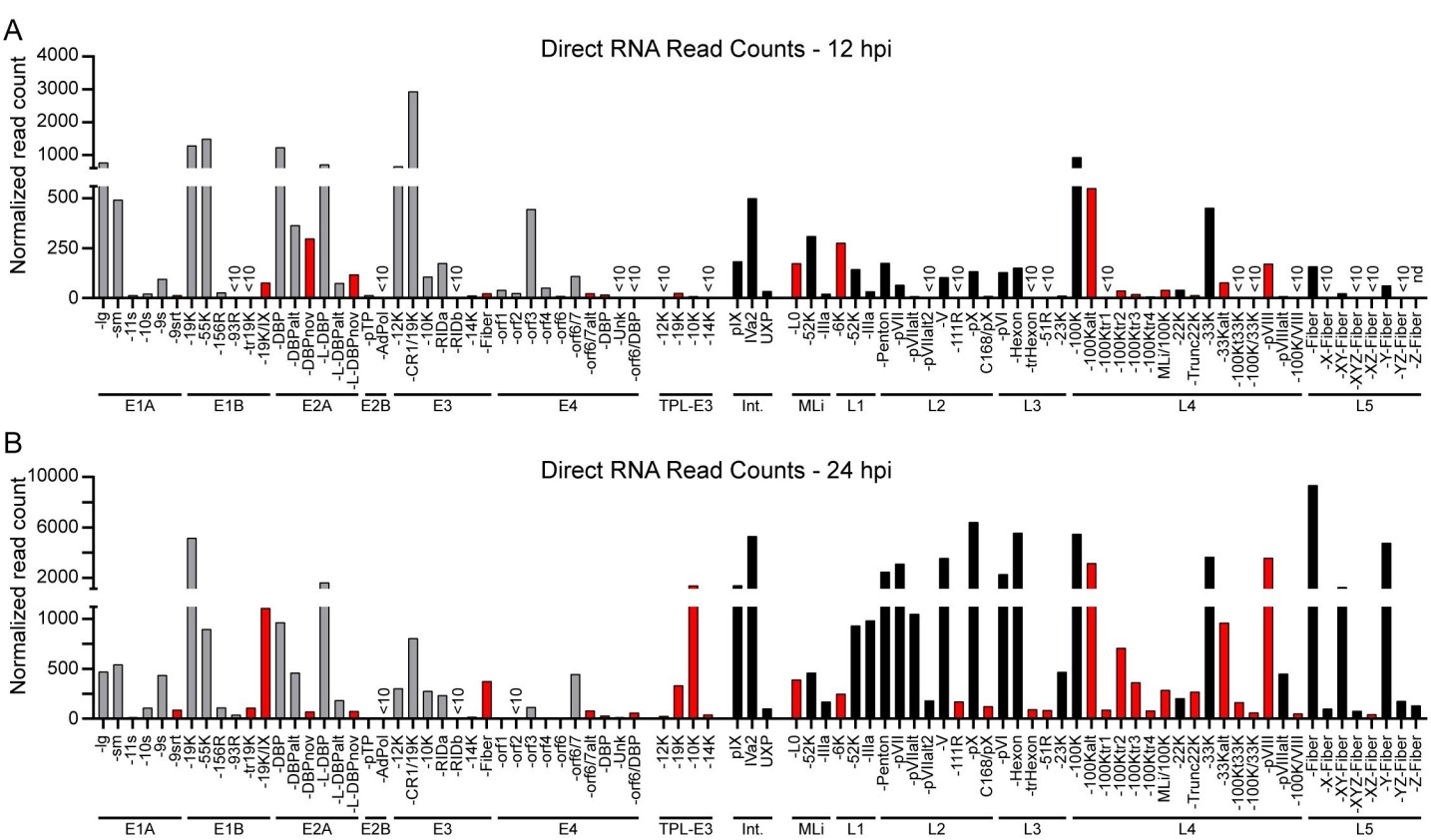

**Fig 3. Direct RNA Sequencing (dRNA-seq) unambiguously distinguishes early and late transcription. (A)** dRNA-seq was performed on polyadenylated RNA from Ad5-infected A549 cells extracted at 12 hours post-infection (hpi). Sequence reads were aligned to the re-annotated transcriptome and filtered to retain only unambiguous primary alignments. Normalized read count indicates the number of RNAs for a particular transcript once normalized to the total number of mappable reads (human plus adenovirus) for the entire sequencing reaction. For all panels, grey bars indicate early genes, black bars indicate late genes, and red bars indicate novel isoforms discovered in this study. Undetectable transcripts (nd) or those with fewer than 10 counts of a particular isoform detected (<10) are indicated. **(B)** Same as in Panel A, but with RNA harvested at 24 hpi.

reproducibly detected over a time-course of infection with Ad5, as well as display differential expression based on the stage of infection.

## The novel fusion transcript between E4orf6 and DBP is expressed, translated, and conserved

We were particularly intrigued by the discovery of transcripts that start within the E4 locus and proceed downstream to the E2A locus, since this event unites viral gene cassettes previously associated with host antiviral antagonism and viral DNA replication, respectively [33]. To be able to splice to downstream E2A acceptors, nascent pre-mRNA must bypass the cleavage and polyadenylation site within the E4 transcriptional unit (**Fig 4A**). Reverse transcription followed by PCR from the E4orf6 region to the downstream DBP exon showed two products of expected size (E4orf6/DBP and E4-Unk) capable of bypassing the E4 polyadenylation site (**Fig 4B**). To ascertain whether any chimeric E4orf6 protein is produced, we leveraged the well-characterized RSA3 antibody that recognizes the N-terminus of E4orf6 [76,77]. Upon immunoblot of A549 cells infected with WT Ad5 for 40 hours the RSA3 antibody detected both the 34 kilodalton full-length E4orf6, as well as E4orf6/7 which results from a splicing and frame-shifting event (**Fig 4C**). Intriguingly, the antibody also detected a

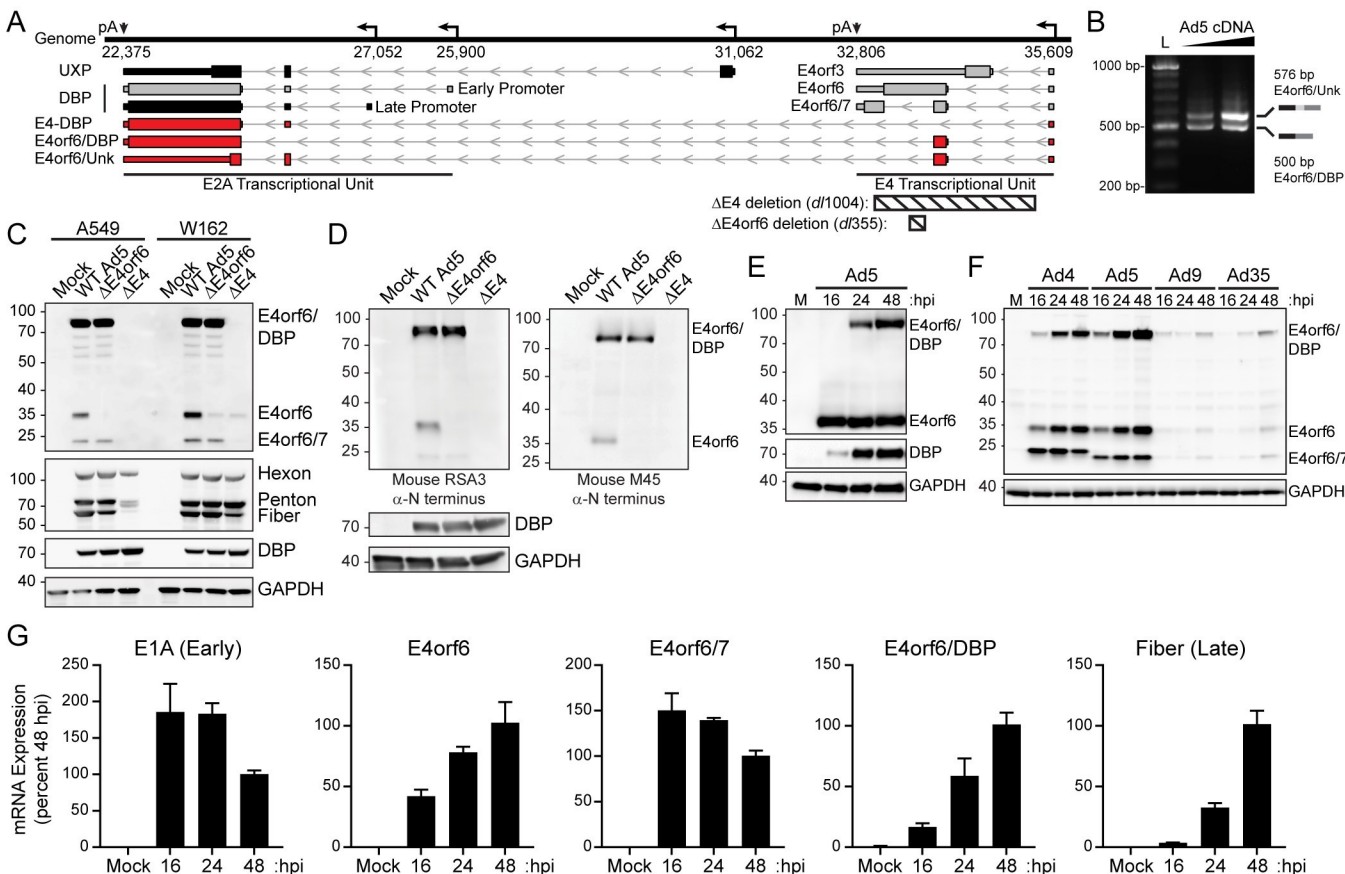

**Fig 4. Novel fusion transcript between E4orf6 and DBP is expressed, translated, and conserved. (A)** Enlarged transcriptome map of Ad5 E4 and E2A transcriptional units. Promoter transcription start sites are indicated with left-facing arrows, and cleavage and polyadenylation sites (pA) labeled with downward facing arrows. Novel E4-derived transcripts that terminate in E2A are highlighted in red. Ad5 mutant viruses *dl*1004 and *dl*355 contain deletions (indicated by hashed boxes) that remove most of the E4 region and splice donors (ΔE4) or a 14-base deletion inside E4orf6 that only abrogates E4orf6 expression (ΔE4orf6). **(B)** Reverse-transcriptase PCR on cDNA derived from Ad5-infection of A549 cells reveals characteristic bands of both E4orf6/Unk and E4orf6/DBP. L denotes DNA ladder and triangle indicates increasing cDNA concentration. **(C)** A549 or W162 cells were uninfected (mock) or infected with WT Ad5, ΔE4orf6, or ΔE4 viruses for 40 hours and proteins detected by immunoblot analysis. When blotting with antisera raised against the N-terminus of E4orf6, a prominent band is detected at the predicted size of E4orf6/DBP. This band is absent during ΔE4 infection and not observed in W162 cells where only the E4 region is provided *in trans*. Kilodalton size markers are shown to the left of each blot. **(D)** Infections and immunoblot analysis were performed as described in panel **C**. Two independently derived anti-N-terminal E4orf6 antibodies (RSA3 and M45) detect E4orf6/DBP. **(E)** Proteins expressed over a time-course of Ad5 infection in A549 were detected by immunoblot analysis at indicated hpi. **(F)** A549 cells were infected with adenoviruses from four different serotypes in a time-course. All tested adenovirus serotypes express a protein corresponding to E4orf6/DBP. **(G)** Quantitative reverse-transcriptase PCR was performed to demonstrate mRNA accumulation of Ad5 E1A (a representative early transcript), Fiber (a representative late transcript), and three E4orf6 containing transcript isoforms. Transcripts were normalized to expression level at 48 hpi and internal HPRT1 housekeeping gene.

robust band around 80 kilodaltons which corresponds to the predicted molecular weight of the N-terminus of E4orf6 fused to the entire open reading frame of the 72 kilodalton DBP. Leveraging the power of AdV genetic knockouts, we showed that a ΔE4orf6 virus (*dl*355 [78]) lacks expression of full-length E4orf6 due to a 14 bp deletion leading to premature translation termination, while retaining the RNA splice donor that allows for expression of both E4orf6/7 as well as the putative E4orf6/DBP. Furthermore, infection with a ΔE4 virus (*dl*1004 [31]) that abrogates all splice donors and acceptors within the E4 region led to loss of the bands corresponding to E4orf6, E4orf6/7, and E4orf6/DBP. The ΔE4 virus is highly defective for viral replication, however most functions can be restored by infecting Vero W162 cells that contain an integrated copy of the E4 region and therefore provide expression of E4 RNAs *in trans* upon infection [79]. Importantly, in human cells splicing of pre-

mRNA must occur *in cis*, and it is very rare to observe *trans*-splicing. Infection of W162 cells with ΔE4orf6 or ΔE4 mutant viruses restored low-level expression of both E4orf6 and E4orf6/7, but ΔE4 virus infection did not show expression of E4orf6/DBP (**Fig 4C**). Finally, to rule out that the detected band was a cross-reactive species, an independently derived anti-N-terminal E4orf6 antibody (M45) was used to recapitulate the finding with the previously described viral mutants (**Fig 4D**). When reviewing the literature for additional examples of uncropped immunoblots utilizing N-terminal E4orf6 antibodies, we noticed that multiple other groups also detected a band corresponding to the molecular weight of E4orf6/DBP [77,80]. Thus, the genetic and biochemical evidence indicates that E4orf6/DBP is produced during Ad5 infection.

We were surprised to see such a prominent band for E4orf6/DBP protein when compared to the equivalently expressed E4orf6 mRNA or more highly expressed E4orf6/7 mRNA. Since our transcriptomic analysis (**Fig 3**) predicted that E4orf6/DBP might possess late gene kinetics, we explored E4orf6/DBP expression over a time-course of infection. Upon infection of A549 cells with WT Ad5 we saw that E4orf6 had reached maximal expression as early as 16 hpi, whereas E4orf6/DBP was first detected at 24 hpi and reached maximal expression at 48 hpi (**Fig 4E**). Similarly, when A549 cells are infected with diverse human AdV serotypes, the E4orf6/DBP protein was detected and increased in expression in a time dependent manner (**Fig 4F**). Finally, when quantitative reverse transcription PCR was performed to asses mRNA expression characteristics of E4orf6, E4orf6/7, and E4orf6/DBP, the E4orf6/DBP transcript had kinetics more similar to a known late gene (Fiber) than a representative early mRNA (E1A) (**Fig 4G**). Thus, E4orf6/DBP is conserved among diverse AdV serotypes, and expressed as a true late gene during infection.

## E4orf6, but not E4orf6/DBP, generates an E3 ubiquitin ligase complex with E1B55K

E4orf6 is most well characterized to bridge the viral targeting receptor E1B55K to a cellular E3 ubiquitin ligase composed of Elongin B, Elongin C, and Cullin5 [21–23]. The region necessary for binding E1B55K has been mapped to the N-terminus of E4orf6, and therefore we considered whether fusion of this 58 amino acid stretch to the disordered N-terminus of DBP could regulate ubiquitin ligase activity (**Fig 5A**). We generated N-terminal flag-tagged versions of both E4orf6 and E4orf6/DBP in mammalian expression constructs. HEK293 cells contain an integrated copy of AdV E1A and E1B such that when E4orf6 is transfected a viable E1B55K/E4orf6 ubiquitin ligase is reconstituted [81]. We verified that expression of E4orf6 alone was sufficient to degrade known ligase targets Rad50 and Mre11, and that addition of the N-terminal flag tag to E4orf6 did not hinder this activity (**Fig 5B**). Expression of flag-E4orf6/DBP led to detection of an RSA3-responsive band at the expected size, but had no effect on stability of Rad50 or Mre11. Furthermore, co-expression of E4orf6 with E4orf6/DBP did not hinder the ability of E4orf6 to form a functional ubiquitin ligase.

Previous studies using monoclonal antibodies against E4orf6 have reported E4orf6 to be present at viral replication centers (VRCs) marked by DBP by indirect immunofluorescence assay [82–84]. In fact, E4orf6 localization to DBP was thought to bring a functional viral ubiquitin ligase to sites of viral DNA replication and RNA transcription [85–87]. Now that we know a major fraction of the E4orf6 signal detected at late times of infection by RSA3 or M45 immunoblot is actually E4orf6/DBP, we revisited this subcellular localization determined by immunofluorescence. In uninfected HEK293 cells, flag-tagged E4orf6 and E4orf6/DBP were both diffusely nucleoplasmic (**Fig 5C**). During infection, DBP-marked replication centers recruit the cellular deubiquitinase USP7 [88]. While flag-E4orf6/DBP

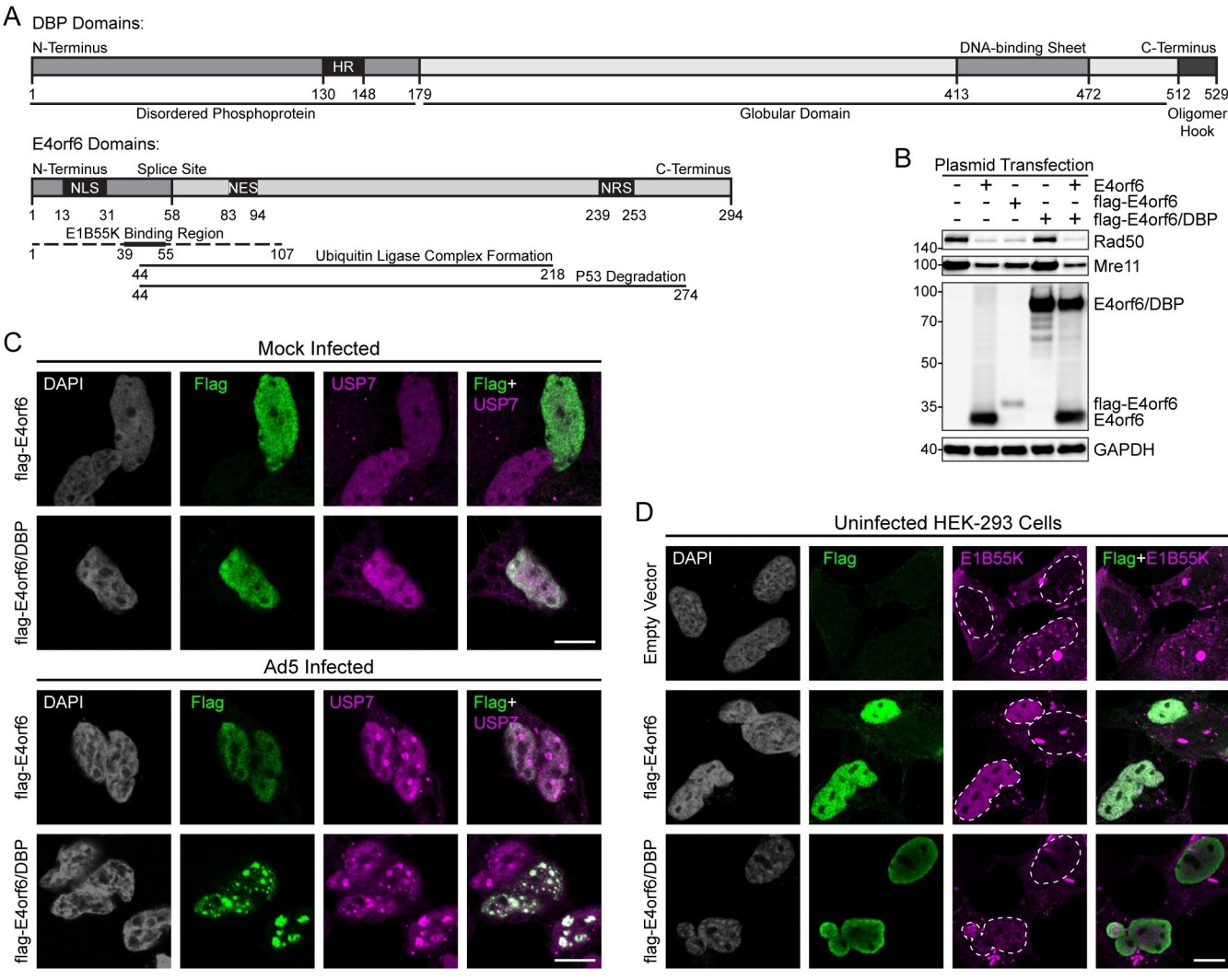

**Fig 5. The E4orf6 protein generates an E3 ubiquitin ligase complex with E1B55K but this is not observed for the E4orf6/DBP fusion. (A)** Domain map of DBP and E4orf6 open reading frames. HR: host range, NLS: nuclear localization sequence, NES: nuclear export sequence, NRS: nuclear retention sequence. **(B)** HEK293 cells were transfected with E4orf6, flag-E4orf6, or flag-E4orf6/DBP and subjected to immunoblot analysis. E4orf6, but not E4orf6/DBP, induces degradation of Mre11 and Rad50 through a complex with E1B55K **(C)** HEK293 cells were transfected with flag-E4orf6 or flag-E4orf6/DBP and then subsequently mock-infected or Ad5-infected for 24 hours. Immunofluorescence was performed for flag tag (green) or cellular USP7 (magenta). Both proteins are nuclear but only E4orf6/DBP localizes to viral replication centers marked by USP7. **(D)** HEK293 cells were transfected with flag-E4orf6 or flag-E4orf6/DBP. Immunofluorescence was performed for flag tag (green) or integrated E1B55K (magenta). Only E4orf6 induces relocalization of E1B55K from cytoplasmic aggresomes to the nucleus. Dashed white lines outline the nuclear periphery. White scale bar denotes 10 μm.

was recruited to USP7-marked VRCs during infection, flag-E4orf6 remained markedly diffuse throughout the nucleus. Another known consequence of the interaction of E4orf6 with E1B55K is the dispersal of E1B55K "aggresomes" present in uninfected HEK293 cells [89]. We confirmed that expression of flag-E4orf6 leads to a relocalization of E1B55K into a diffuse nuclear pattern within transfected cells, however E4orf6/DBP was did not reorganize E1B55K (**Fig 5D**). Thus, E4orf6/DBP has different cellular localizations compared to E4orf6, and does not bind to E1B55K or participate in ubiquitin ligase activities. Furthermore, our findings recontextualize previous knowledge about the localization of E4orf6 during infection.

## Loss of E4orf6/DBP has minimal impact on viral genome replication or protein expression

Both E4orf6 and DBP are critical for AdV replication, and therefore any strategy to delete fragments of E4orf6/DBP would have substantial off-target effects. Instead, we opted to create a silent mutation in E4orf6 that abrogates the ability of the cellular U1 snRNP from recognizing the RNA splice donor necessary for creation of both E4orf6/7 and E4orf6/DBP [90] (**Fig 6A**). While this strategy involved creating a double knockout, prior work has demonstrated that E4orf6/7 is dispensable for viral replication in cell culture due to redundancy with AdV E1A [78,91]. Importantly, prior research studying E4orf6/7 deleted viruses has employed frameshifting deletions within the downstream E4orf6/7 exon, thus the E4orf6 splice donor (and therefore E4orf6/DBP expression) remained intact. To confirm loss of E4orf6/DBP expression, we performed time-course infections in A549 cells using WT Ad5, the E4orf6/7;E4orf6/DBP double KO (ΔSS), and the previously used *dl*355 ΔE4orf6 viruses (**Fig 6B**). Loss of E4orf6 resulted in the previously reported decrease of AdV late protein expression (Hexon, Penton, and Fiber) at 24 hpi, and a partial loss of degradation of the ubiquitin ligase substrate Rad50. In contrast, the ΔSS virus showed a complete loss of both E4orf6/7 and E4orf6/DBP, was fully competent for degrading Rad50, and showed no defect in accumulation of viral early proteins (E4orf6 and DBP) or viral late proteins (Hexon, Penton and Fiber). To complement specific loss of E4orf6/DBP we devised a retroviral system to generate cell lines which express adenoviral proteins under the control of the endogenous viral E4 promoter (**S2A Fig**). This system can be used to express either flag-E4orf6 or flag-E4orf6/DBP only in cells infected by AdV and expressing the viral E1A transcription factor. Using this orthogonal system, we corroborated our finding that E4orf6/DBP co-localized with VRCs, whereas E4orf6 remained broadly nucleoplasmic upon infection (**S2B and S2C Fig**). When these A549:E4orf6/DBP cells were infected with WT Ad5 or the ΔSS virus we again observed no defect in the expression of viral early or late proteins (**Fig 6C**). Retroviral expression of flag-E4orf6/DBP was expressed with similar kinetics and abundance to endogenous E4orf6/DBP.

Besides the formation of viral replication centers, DBP is a single-stranded DNA binding protein essential for replication of AdV DNA genomes [27]. We next asked whether E4orf6/DBP might modulate either of these functions of DBP. To ascertain whether the loss of E4orf6/DBP affected genome replication, we infected A549 cells with WT Ad5 or the ΔSS virus, and isolated genomic DNA to perform quantitative PCR over a time-course of infection (**Fig 6D**). While there was a slight defect (<2-fold) of viral genome accumulation with the ΔSS virus at 48 hpi, there was no significant decrease in DNA replication at early times post-infection. We next utilized the ΔSS virus to assay for localization of endogenous E4orf6 proteins and viral replication centers. Performing indirect immunofluorescence with the monoclonal anti-E4orf6 antibody (RSA3) during WT Ad5 infection, we confirmed previous reports from the literature that the E4orf6 signal co-localized with viral replication centers (**Fig 6E**). Of note, infection with the *dl*355 virus entirely lacking E4orf6 still resulted in RSA3 staining at VRCs (**Fig 6E**). Within our 24 hpi snapshot we observed multiple cells that show the earliest signs of infection (i.e., small viral replication centers marked by USP7; closed arrow) indicating that these cells had not progressed to the late stage of infection, yet early expression of E4orf6 at VRCs was not detected. These data support a model where VRC localized E4orf6 signal was likely E4orf6/DBP expressed with late kinetics, and not E4orf6. In contrast, the ΔSS virus that expresses wildtype-levels of E4orf6, but not E4orf6/DBP, only had diffuse nuclear staining of RSA3 signal and lacked the intense staining at viral replication centers. In addition, infection with *dl*356 (which lacks E4orf6/7 [78]) and *dl*1002 (which lacks both E4orf6 and 6/7, but retains E4orf6/DBP due to the mutation being downstream of the splice donor [31]) both

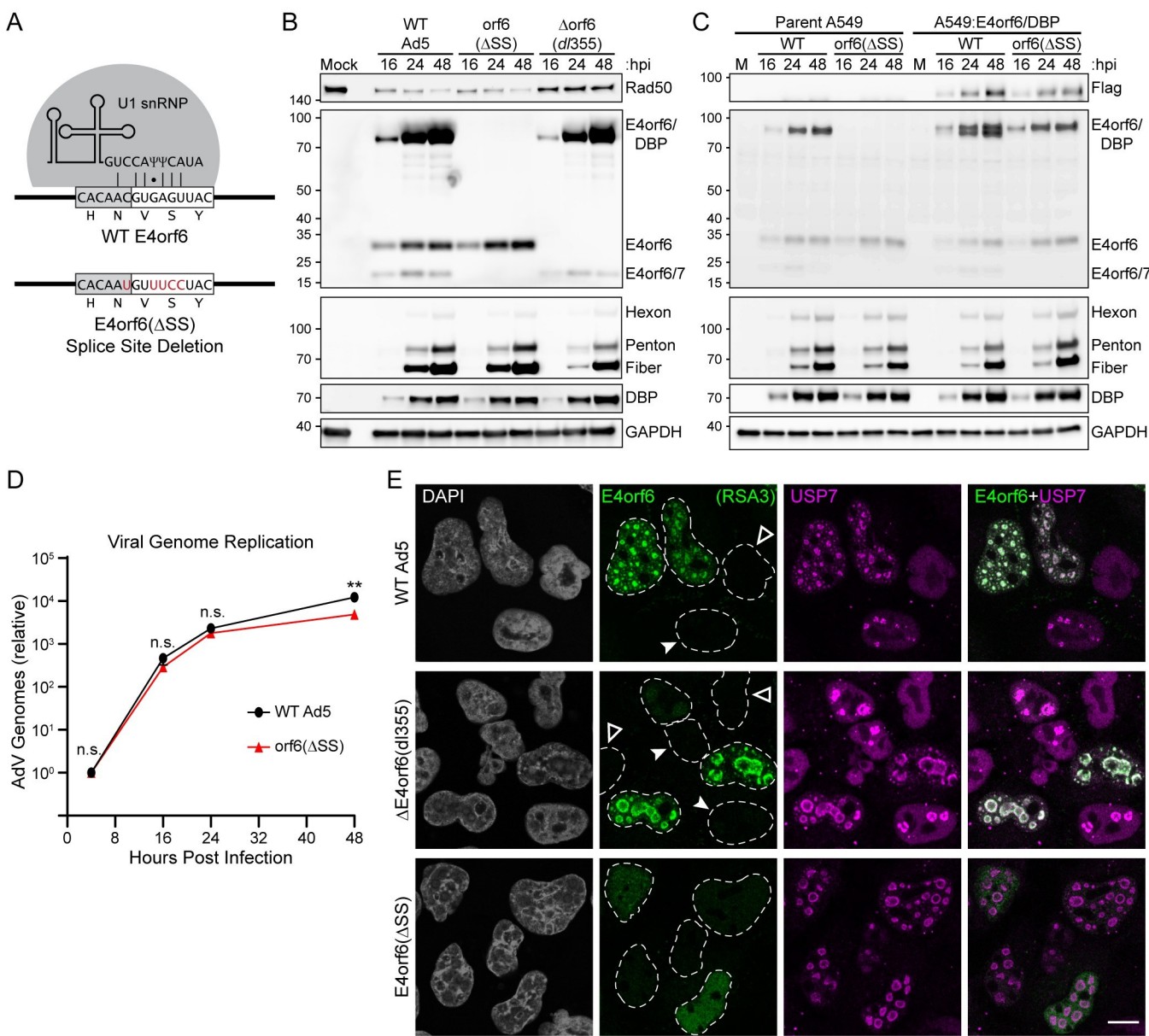

**Fig 6. Loss of E4orf6/DBP has minimal impact on viral genome replication or protein expression. (A)** E4orf6 splice donor is recognized by base-pairing to the cellular U1 snRNP. Silent mutations shown in red abrogate downstream splicing and were used to create the E4orf6/7, E4orf6/DBP double knockout virus (E4orf6ΔSS). **(B)** A549 cells were infected with WT Ad5, E4orf6ΔSS, or ΔE4orf6 (*dl*355) and harvested over a time-course of infection (hpi, hours post-infection). Immunoblot analysis was performed with antibodies to detect the indicated viral and cellular proteins. Kilodalton size markers are shown to the left of each blot. **(C)** Immunoblot analysis of a time-course infection was performed as in panel **B**. Cell lines were parent A549 or A549 transduced to express flag-E4orf6/DBP under control of the adenoviral E4 promoter. **(D)** A549 cells were infected with WT Ad5 or E4orf6ΔSS in biological triplicate. Adenoviral genome copy number was determined by qPCR and normalized to the amount of input genomes at 4 hpi. Significance was determined by unpaired, two-tailed t-test (n.s., not significant; $^{**}$, p-value<0.01). **(E)** A549 cells were infected with WT Ad5, ΔE4orf6, or E4orf6ΔSS for 24 hours before immunofluorescence was performed. Cells were stained with antibodies against E4orf6 N terminal domain (RSA3, green) or cellular USP7 as a marker of viral replication centers (magenta). Dashed white lines outline the nuclear periphery. Open white arrowheads denote uninfected cells with diffuse USP7 and no viral staining, while closed arrowheads denote infected cells that lack E4orf6 staining but show USP7 at viral replication centers. White scale bar denotes 10 μm.

retain prominent staining of RSA3 at VRCs (**S3 Fig**). These data corroborate our overexpression studies and confirm that endogenous E4orf6 does not localize to viral replication centers. Furthermore, while loss of E4orf6/DBP had only minimal effect on viral genome replication or protein expression, we noticed a striking change in the appearance of VRCs.

## Loss of E4orf6/DBP leads to altered viral replication center morphology and small plaque phenotypes

Nuclear-replicating DNA viruses often compartmentalize viral DNA replication and other processes into membraneless organelles termed viral replication centers (VRCs) [37,92]. Early work characterized these sites as small, spherical bodies that possess liquid-like characteristics of being able to fuse and split apart. However, recent studies have expanded upon the standard model of VRCs and demonstrate dynamic morphological changes that occur upon the onset of late gene expression. While early replication centers are round or crescent shaped [93], these VRCs appear to break apart and become more diffuse before ultimately coalescing in ring-like structures within the center of the nucleus [94] (**Fig 7A**). This biphasic model of VRC morphology correlates with changes in the spatial location and rate of genome synthesis, and ultimately leads to the accumulation of viral DNA in virus-induced post-replicative (ViPR)

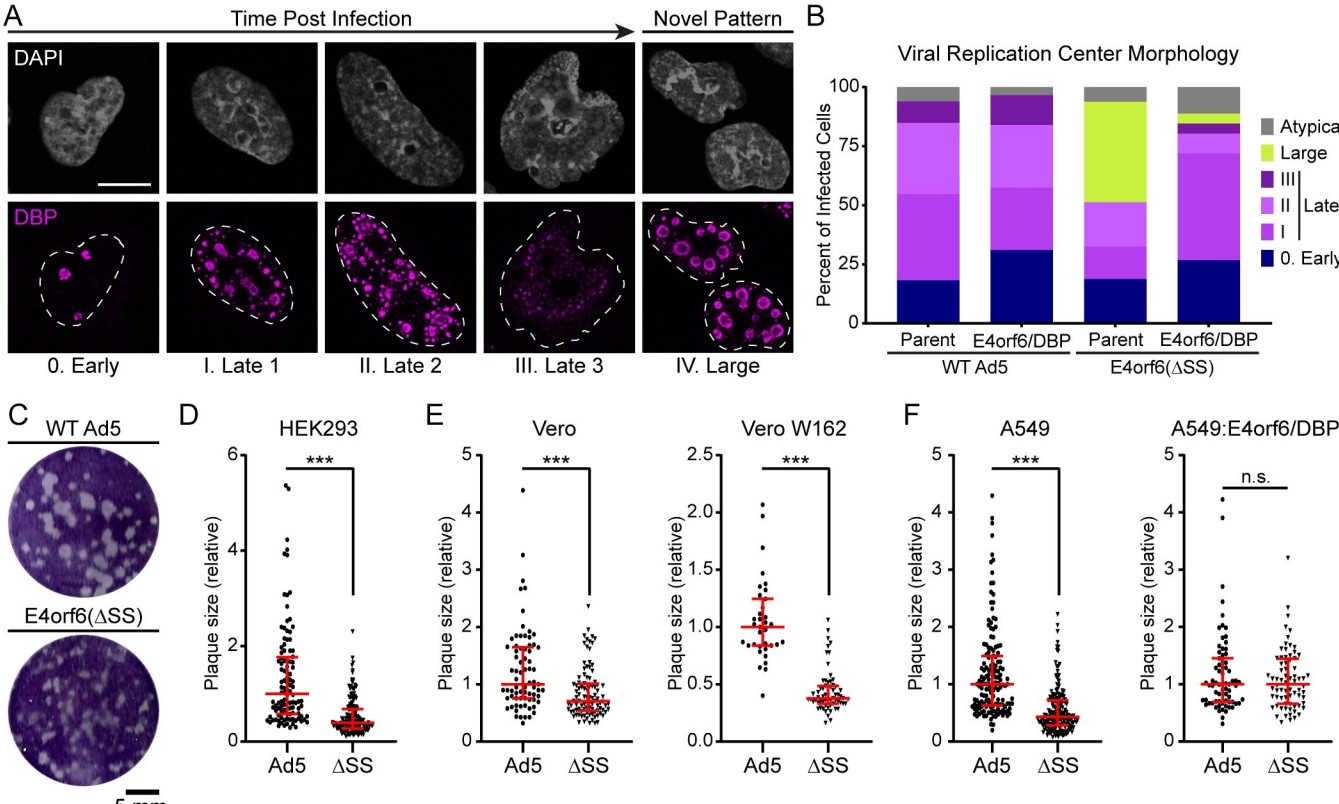

**Fig 7. Loss of E4orf6/DBP leads to altered viral replication center morphology and small plaque phenotypes.** **(A)** Wildtype adenovirus infection leads to temporal progression of viral replication center (VRC) morphology progressing from early stage (0. Early) through three distinct stages of late replication centers (Late 1–3). A novel, large replication center morphology was seen with infection of E4orf6ΔSS virus. Representative immunofluorescence images for each stage are shown by staining A549 cells with VRC marker DBP (magenta) and DAPI for DNA (grey). **(B)** Parent A549 (Parent) or A549 cells expressing E4p-flag-E4orf6/DBP (E4orf6/DBP) were infected with WT Ad5 or E4orf6ΔSS virus for 24 hours. Viral replication centers were stained for DBP, and morphology was scored in a blinded manner using the key provided in panel **A**. The percentage of infected cells demonstrating each replication center morphology is shown as bar chart. Data are representative of three independent experiments. **(C)** HEK293 cells were infected with limiting dilution of WT Ad5 or E4orf6ΔSS for six days to allow the formation of plaques. Plaque formation was negative stained with crystal violet and imaged. Scale bar shows 5 mm. **(D)** WT Ad5 or E4orf6ΔSS (ΔSS) plaque size in HEK293 cells was quantified with ImageJ and normalized to the median plaque size in WT infection. Individual plaques are plotted as a single point. **(E)** Plaque size was quantified in Vero cells or Vero-W162 cells. W162 cells express the adenovirus E4 region *in trans* and complement the loss of E4orf6/7 in the ΔSS virus. **(F)** Plaque size was quantified in A549 cells or A549:E4orf6/DBP cells. The A549:E4orf6/DBP cells express E4orf6/DBP under control of the E4 promoter *in trans*, and thus complement the loss of E4orf6/DBP in the ΔSS virus. For all plaque assays, red error bars denote median and interquartile range. Statistical significance was performed using non-parametric Mann-Whitney t-test (n.s., not significant; ***, p-value<0.001).

bodies [95,96]. While the progression through the distinct VRC morphologies is correlated with time post-infection, in any given snapshot of infection multiple classes of VRCs can be detected due to the asynchronous nature of viral replication on a cell-by-cell basis. Strikingly, when we visualized viral infection by staining with DBP (**Fig 7A**) or USP7 (**Fig 6E**) during infection with the ΔSS virus, we noticed a large fraction of cells possessed a novel VRC morphology characterized by exceptionally large, round or donut-shaped bodies. This novel VRC morphology was not seen with the loss of only E4orf6/7 (**S3 Fig**). To quantify VRC morphology we performed phenotype analysis by confocal microscopy for A549 cells infected by WT Ad5 or ΔSS virus for 24 hours within three independent replicates. All infected cells were categorized in a blinded fashion into early replication centers (0. Early), one of three distinct late-stage VRC morphologies (I., early VRCs with numerous small foci; II., more complete breakup of VRCs into a multitude of medium bodies; or III., ring-like patterns with ViPR bodies), the novel phenotype (IV., large), or a sixth category of minority variants that defied typical classification (Atypical). At this time point greater than 60% of WT-infected cells had progressed to the late stage of VRC morphology (**Fig 7B**). While a similar fraction of ΔSS-infected cells were infected and reached the early stage of VRC morphology, greater than 40% of cells now possessed the novel large VRC phenotype. When WT or ΔSS virus were used to infect A549: E4orf6/DBP cells that complement E4orf6/DBP under control of the endogenous viral promoter, the amount of ΔSS-infected cells displaying the novel VRC phenotype was dramatically reduced and instead displayed normal late-stage VRC morphologies (**Fig 7B**). This implies that while E4orf6/DBP is only a small fraction of the total pool of DBP protein isoforms, it has an outsized impact on the evolution of VRC morphology within infected cells.

Despite the striking change in VRC morphology, the ΔSS virus has minimal effects on genome accumulation or viral protein expression during a single cycle of infection (**Fig 6**). We next asked what effect the loss of E4orf6/DBP might have on viral replication during multiple cycles of infection. One measure of viral spread over multiple cycles of infection is the ability to form infectious plaques in a monolayer of cells. We performed serial dilutions of WT Ad5 and ΔSS virus on HEK293 and allowed plaques to form underneath agarose for exactly five days (**Fig 7C**). Despite the similar number of total plaques, the plaques formed by WT Ad5-infection were much larger than those formed by ΔSS-infection in the same amount of time, as could be quantified by ImageJ software (**Fig 7D**). Importantly, a virus that only lacks E4orf6/7 formed plaques that were not significantly different than WT infection of HEK293 cells (**S4 Fig**). When we performed the same assay in Vero cells, or Vero W162 cells that can complement the lack of E4orf6/7 but not E4orf6/DBP during ΔSS-infection, we again quantified a significant reduction in plaque size when comparing the WT Ad5 to ΔSS virus (**Fig 7E**). Finally, while a significant disparity in plaque size could once again be quantified in parental A549 cells, plaques formed on A549:E4orf6/DBP cells showed no significant difference in plaque size (**Fig 7F**). Therefore, these data suggest that E4orf6/DBP contributes to the cell-to-cell spread or cell killing during a multi-cycle viral infection.

## Discussion

DNA viruses encode large amounts of information in compact genomes through alternative splicing, overlapping transcripts, and transcription from both strands of the genome. Here we integrate short-read cDNA sequencing and long-read direct RNA sequencing to re-annotate both the Ad5 DNA genome and RNA transcriptome. Using high quality and high depth short-read sequencing, we were able to detect SNPs within the transcribed regions of the genome approaching 100% penetrance, indicating that these sites were likely present in the genome and not due to RNA editing or modifications. We recapitulated the known TSS and CPAS

sites throughout the Ad5 genome, and annotated novel splicing events within the viral transcriptome. Of these 35 novel RNAs, 15 are likely to encode for altered ORFs including multiple fusion transcripts that span transcriptional units previously thought to be distinct. Of these, we focused on one transcript, E4orf6/DBP, which we show produces a novel conserved protein with late gene kinetics. Study of E4orf6/DBP recontextualized our knowledge of the critical AdV effector protein E4orf6, as well as revealed a unique role for this chimeric viral protein in regulating viral replication center morphology and cell-to-cell spread. Overall, we have provided a more comprehensive annotation of a complex viral transcriptome that highlights multiple new gene products for future study.

Using RNA-seq data to call SNPs in viral DNA genomes represents a compelling approach to identify SNPs, since high quality short-read sequencing data sets already exist for many DNA viruses [75,97–100]. While half of the SNPs we called were synonymous or in non-coding regions, missense mutations have the potential to change the coding sequence of protein amino acids in meaningful ways. In addition, while SNPs are often tolerated during alignment of RNA-seq data, annotation of the correct primary amino acid sequence is critical for downstream analysis of mass spectrometry data [101]. These SNPs were validated by DNA-sequencing of a common Ad5 plasmid used in viral mutagenesis by multiple laboratories, highlighting a need to revisit existing reference genomes for common DNA viruses. Of note, Westergren-Jakobssen *et al.*, also discovered SNPs by directly sequencing the DNA of Ad2 [62], however none of these SNPs overlapped with the nucleotide polymorphisms we detected in the related Ad5.

Previous studies of AdV transcription detected numerous splice sites beyond those employed by known isoforms. However, the constraints of short-read sequencing precluded proper assembly of these sites into full-length transcripts [44,54,73]. Furthermore, targeted expression analysis over a time-course of infection was limited to already known transcripts [55]. In the past two years, two groups have independently used direct RNA sequencing on the Oxford Nanopore Technology platform to profile adenoviral RNA expression of two related serotypes, Ad2 and Ad5 [61,62]. While both studies highlighted an exceptional diversity of viral transcripts produced during infection, many novel transcripts were expressed at very low levels or only seen very late after infection. By combining high-depth short-read sequencing, direct RNA long-read sequencing, and stringent cutoffs, we can now report high confidence novel transcripts as well as show regulated expression over a time-course of infection. Nearly all of the high confidence novel transcripts we report on were also detected by both prior studies, including the E4orf6/DBP transcript we have focused on (**S2 Table**). We have also added ORF predictions to previously detected splice sites, such as L2-111R, X-Z-Fiber, and the pVIII ORF derived from splicing directly from the tripartite leader to the L4-33K splice acceptor. While this last site was previously predicted to lead to the expression of a small 42 amino acid ORF [54], we propose that this transcript instead primarily encodes for pVIII with a small upstream ORF, as it is over five times as abundant as the canonical pVIII spliced RNA.

Of the novel transcripts we have so far detected, all of them display delayed late kinetics during infection. Of particular interest is the transcript encoding for a fusion event between the E4 transcriptional unit and the E2 transcriptional unit. This transcript, E4orf6/DBP, would have to skip the canonical E4 cleavage and polyadenylation site for the pre-mRNA to progress downstream to DBP for splicing. The three transcripts displaying this pattern, including E4-promoter driven DBP and frameshifted E4-Unk, are all much more abundant during the late phase of infection even though canonical E4 transcripts are expressed early. Other transcripts displaying readthrough of polyadenylation include an E1A-9s mRNA, E3-driven Fiber, as well as nearly all canonical late transcripts. In fact, Fiber driven by the major late promoter must bypass seven actionable polyadenylation sites before revealing the 3' splice acceptor. It will be interesting to see the extent to which differential polyadenylation is regulated during

the life cycle of the virus, as has been previously reported for herpes simplex virus [58,99,102]. Alternatively, AdV late transcripts might employ recursive splicing to lessen the need for long introns [103].

Of the 20 novel Ad5 protein isoforms we detected within our reannotated transcriptome, we have been able to provide mass spectrometry confirmation for 12 candidates (60%). For the remaining eight proteins absence of evidence is not evidence of absence, since mass spectrometry requires peptide fragments of ideal mass and charge for detection. By exploring just one of these novel proteins, we have been able to provide genetic and biochemical evidence for the expression, evolutionary conservation, and function of E4orf6/DBP. While it appears that E4orf6/DBP regulates both the morphology of viral replication centers and cell-to-cell spread, the molecular mechanism of these functions remains to be determined. The DBP protein is known to bind single-stranded DNA and oligomerize to form chains upon replicating DNA [27]. While loss of E4orf6/DBP did not affect viral DNA accumulation, it is tempting to speculate that addition of the E4orf6 fragment to the N-terminus of DBP might affect the localization of viral DNA replication or the ability of DBP to form oligomers. Prior to this study, the most recently discovered protein of Ad5, UXP, was shown to be generated from an mRNA that is co-terminal with DBP, translated in an alternate reading frame to DBP, localized to DBP-marked replication centers, and led to altered VRC morphology in its absence [48,49]. The complete protein complement of viral replication centers is unknown, and it is likely that E4orf6/DBP might alter viral or cellular protein components that localize to these compartments. These interactions might be modulated by affecting the N-terminus of DBP which contains substantial intrinsic disorder and is post-translationally modified by phosphorylation, both of which are known to mediate protein-protein interactions. Finally, recent evidence has provided support for a model where AdV particles are packaged in proximity to VRCs [104]. Changes in AdV particle production could result from modification of these VRCs or access to the viral DNA, and these alterations might impact cell-to-cell spread.

A complication arising from the study of viral pathogens is that otherwise simple knockouts or mutations might alter the coding potential of nearby or anti-sense coding units [105]. In our characterization of E4orf6/DBP we have discovered that the previously understood localization of E4orf6 to VRCs was incorrect due to the presence of a previously undiscovered protein being recognized by the same monoclonal antibody. Importantly, future research should identify whether the known functions of existing viral ORFs can be explained, at least in part, by the presence of these novel isoforms. We have provided an example of how modern tools in RNA sequencing and mass spectrometry can offer a more complete reference and framework for undertaking research on a given complex pathogen. In doing so, we have added new depth to a common virus that has been exceptionally well-characterized for many decades.

## Methods

### Cell culture

All cell lines were obtained from American Type Culture Collection (ATCC) and cultured at 37°C and 5% CO2. A549 cells (ATCC CCL-185) were maintained in Ham's F-12K medium (Gibco, 21127–022) supplemented with 10% v/v FBS (VWR, 89510–186) and 1% v/v Pen/Strep (100 U/ml of penicillin, 100 μg/ml of streptomycin, Gibco, 15140–122). HEK293 cells (ATCC CRL-1573), Vero cells (ATCC CCL-81), and W162 cells (Kind gift of G. Ketner) were grown in DMEM (Corning, 10-013-CV) supplemented with 10% v/v FBS and 1% v/v Pen/Strep. All cell lines tested negative for mycoplasma infection and were routinely tested afterwards using the LookOut Mycoplasma PCR Detection Kit (Sigma-Aldrich).

## Viral infections

Adenovirus serotype 5 (Ad5), Ad4, Ad9, and Ad35 were originally purchased from ATCC. Ad5 ΔE4orf6 (*dl*355), ΔE4orf6/7 (*dl*356), ΔE4orf6,6/7 (*dl*1002), E4in6,6/7 and ΔE4 (*dl*1004) were previously described and obtained from D. Ornelles [31,78,106]. Construction of the E4orf6 splice site deleted virus (ΔSS) was performed as previously described [107,108], using Ad5 cloning plasmids that were a kind gift from P. Hearing. In brief, we used site-directed mutagenesis to silently mutate the E4orf6 splice donor in the plasmid pBS-E4. Subsequently, a linear HindIII/KpnI fragment containing the ΔSS mutation was electroporated into BJ5183 E. coli cells (Fisher Scientific, 50-125-019) with the ClaI linearized pTG3602*in*Orf3 Ad5 plasmid. Overnight colonies were screened for insertion of the splice-site deletion, expanded in DH5α E. coli, and maxi-prepped. Insertion of the splice-site deletion repair of the E4orf3 insert were verified with Sanger Sequencing (GeneWiz), and integrity of the viral plasmid was compared by restriction-fragment mapping to wildtype Ad5 (pTG3602). All viruses were expanded on HEK293 cells, purified using two sequential rounds of ultracentrifugation in CsCl gradients, and stored in 40% v/v glycerol at -20˚C (short term) or -80˚C (long term). Viral stock titer was determined on HEK293 cells by plaque assay, and all subsequent infections were performed at a multiplicity of infection (MOI) of 10 PFU/cell. Cells were infected at 80–90% confluent monolayers by incubation with diluted virus in a minimal volume of low serum (2%) F-12K for two hours. After infection viral inoculum was removed by vacuum and full serum growth media was replaced for the duration of the experiment.

## Immunoblotting and antibodies

For immunoblotting analysis protein samples were prepared by directly lysing cells in lithium dodecyl sulfate (LDS) loading buffer (NuPage) supplemented with 1% beta-mercaptoethanol (BME) and boiled at 95˚C for 10 min. Equal volumes of protein lysate were separated by SDS-PAGE in MOPS buffer (Invitrogen) before being transferred onto a nitrocellulose membrane (Millipore) at 35 V for 90 minutes in 20% methanol solution. Membranes were stained with Ponceau to confirm equal loading and blocked in 5% w/v non-fat milk in TBST supplemented with 0.05% w/v sodium azide. Membranes were incubated with primary antibodies in milk overnight, washed for three times in TBST, incubated with HRP-conjugated secondary (Jackson Laboratories) for 1 h and washed an additional three times in TBST. Proteins were visualized with Pierce ECL or Pico Western Blotting Substrate (Thermo Scientific) and detected using a Syngene G-Box. Images were processed and assembled in Adobe Photoshop and Illustrator CC 2022. Antibodies against adenovirus proteins include rabbit polyclonal against adenovirus Hexon, Penton, and Fiber (Gift from J. Wilson, WB 1:10,000), mouse anti-DBP (Gift from A. Levine, Clone: B6-8, WB 1:1000, IF 1:400), and two mouse anti-E4orf6 (Gift from P. Hearing, Clone: RSA3 or M45, WB 1:400, IF 1:400). Antibodies against cellular proteins include RAD50 (GeneTex GTX70228, WB 1:1000), MRE11 (Novus NB100–142, WB 1:1000), USP7 (Bethyl Laboratories A300–033A, IF 1:500), and GAPDH (GeneTex GTX100118, WB 1:30,000).

## Indirect immunofluorescence assay

Cells were grown on glass coverslips in 24-well plates, mock-infected or infected with virus for the appropriate time, washed twice with PBS and then fixed in 4% w/v Paraformaldehyde for 15 minutes. Cells were permeabilized with 0.5% v/v Triton-X in PBS for 10 mins, and blocked in 3% w/v BSA in PBS (+ 0.05% w/v sodium azide) for one hour. Primary antibody dilutions were added to coverslips in 3% w/v BSA in PBS (+ 0.05% w/v sodium azide) for 1 h, washed with 3% BSA in PBS three times, followed by secondary antibodies and 4,6-diamidino-

2-phenylindole (DAPI) for one hour. Secondary antibodies were used at 1:500 dilution and conjugated to Alexa-Fluor 488 (Invitrogen A-11001 or A-11008), or Alexa-Fluor 568 (Invitrogen A-11004 or A-11011) in anti-mouse or anti-rabbit. Coverslips were mounted onto glass slides using ProLong Gold Antifade Reagent (Cell Signaling Technologies). Immunofluorescence was visualized using a Zeiss LSM 710 Confocal microscope (Cell and Developmental Microscopy Core at UPenn) and ZEN 2011 software. Images were processed in FIJI (v1.53c) and assembled in Adobe Photoshop and Illustrator CC 2022.

## RNA Isolation, DNA Isolation, and Quantitative PCR

Total RNA was isolated from cells by either TRIzol extraction (Thermo Fisher) or RNeasy Micro kit (Qiagen), following manufacturer protocols. RNA was treated with RNase-free DNase I (Qiagen), either on-column or after ethanol precipitation. To test quality, RNA was converted to complementary DNA (cDNA) using 1 µg of input RNA in the High-Capacity RNA-to-cDNA kit (Thermo Fisher). Total DNA was isolated using the PureLink Genomic DNA kit (Invitrogen). Quantitative PCR was performed using the standard protocol for SYBR Green reagents (Thermo Fisher) in a QuantStudio 7 Flex Real-Time PCR System (Applied Biosystems). Primers used included E1A (Fp: GTACCGGAGGTGATCGATCTTA, Rp: TCAGGCTCAGGTTCAGACA), E4orf6 (Fp: TTTACTGGAAATATGACTACGTC, Rp: AAGACCTCGCACGTAACT), E4orf6/7 (Fp: GTAGGGATCGTCTACCTCCTTT, Rp: AGAAGTCCACGCGTTGTG), E4orf6/DBP (Fp: GTAGGGATCGTCTACCTCCTTT, Rp: CTGGCCATTTCCTTCTCGTTG), Fiber (Fp: GAAAGGCGTCTAACCAGTCA, Rp: AAAGGCACAGTTGGAGGAC), DBP (Fp: GCCATTGCGCCCAAGAAGAA, Rp: CTGTCCACGATTACCTCTGGTGAT), HPRT1 (Fp: TGACACTGGCAAAACAATGCA, Rp: GGTCCTTTTCACCAGCAAGCT), and Tubulin (Fp: CCAGATGCCAAGTGACAA GAC, Rp: GAGTGAGTGACAAGAGAAGCC).

## Plaque assay

Viral particles were serially diluted in DMEM supplemented with 2% v/v FBS and 1% v/v Pen/Strep to infect confluent monolayers of HEK293 cells, Vero cells, or A549 cells seeded in 6-well plates. After incubation for 2 h at 37°C, the infection media was removed, and cells were overlaid with DMEM containing 0.45% w/v SeaPlaque agarose (Lonza) in addition to 2% v/v FBS and 1% v/v Pen/Strep. Between 5 to 6 days post-infection plaques were fixed with 10% (wt/vol) trichloroacetic acid was added to each well for 30 min at room temperature. Plaques were stained using 1% w/v crystal violet in 50% v/v ethanol. Plaque assays were scanned and resulting images were processed in FIJI (v1.53c) to determine plaque area. Plaque size in arbitrary units were normalized to the median plaque size from wildtype Ad5 infection in a given cell line.

## Proteomics data analysis

Biological triplicate cells were lysed with 8M urea and sonicated in a Diagenode Biorupter bath sonicator at 4°C on high 30 seconds on and 30 seconds off for 5 minutes. The cells were centrifuged at 10,000xg for 10 minutes at 4°C and the lysates transferred to a new tube. The lysate was then reduced with 5mM dithiothreitol (DTT) for 1 hour at RT and alkylated using 10mM iodoacetamide (IAM) for 45 minutes in the dark and 15 minutes in the light, followed by trypsin at a 1:50 ratio overnight at RT. The following day 200 µg of peptides for each time point and bio-replicate were cleaned using Hamilton spin tip columns. Samples were separated on a standard linear 80-minute gradient from 5% to 60%, using standard proteomics buffers of 0.1% formic acid in aqueous and 0.1% formic acid in 80% acetonitrile (ACN) using a Thermo

Dionex 3000 LC. Samples were quantified using a Thermo Fusion MS instrument and batch randomized to account for instrument variation. The PRM method was designed with the MS1 having a window of 350-1200m/z, resolution of 60K, AGC target of 100% and MIT of 50ms. the tSIM MS$^2$ scan having an AGC target of 200% and MIT of 54ms, resolution of 30K and HCD fragmentation of 28%.

Thermo raw files were imported into Skyline using 2 missed cleavages with a minimum length of 5 and a maximum length of 30 amino acids. Fragments required 4 or more product ions within 5 minutes of the predicted RT window and a mass tolerance of 10ppm or less to be considered for evaluation. Peptides were manually evaluated with the requirement of 5 or more overlapping fragments to be considered a "real" identification in comparison to mock.

Data from *Dybas et al.* [74] was downloaded from the PRIDE database and searched using both Sequest and Byonic sequentially within the Thermo Proteome Discoverer 2.4 platform [109,110]. Using both the new Adenovirus fasta with the previously used human Uniprot fasta database searched at 10ppm MS$^1$ precursor tolerance and 0.02 Da fragment search tolerance for Sequest and 10ppm fragment search tolerance for Byonic. Both search programs used a missed cleavage of 2 and fixed modifications for carbamidomethylation for cysteines and variable modifications for methionine oxidation and N-terminal acetylation. Each search had an independently applied 1% FDR for identified spectra followed by a protein level 1% FDR. Manual validation of peptides was performed with Proteome Discoverer software and visualized through the Coon lab interactive peptide spectral annotator (IPSA) [111].

## Illumina sequencing and mapping

Total RNA from three biological replicates of A549 cells infected with Ad5 for 24 hours were sent to Genewiz for preparation into strand-specific RNA-Seq libraries. Libraries were then run spread over three lanes of an Illumina HiSeq 2500 using a 150bp paired-end protocol. Raw reads were mapped to the GRCh37/hg19 genome assembly and the Ad5 genome using the RNA-seq aligner GSNAP [68] (version 2019-09-12). The algorithm was given known human gene models provided by GENCODE (release_27_hg19) to achieve higher mapping accuracy. We used R package ggplot2 for visualization. Downstream analysis and visualization was done using deepTools2 [112]. Splice junctions were extracted using regtools [113] and visualized in Integrative Genomics Viewer [114].

## Variant calling and whole-plasmid sequencing

Illumina RNA-seq reads were aligned to the Ad5 genome obtained from NCBI (https://www.ncbi.nlm.nih.gov/nuccore/AC_000008) using GSNAP [68]. To identify variants such as single nucleotide polymorphisms (SNPs) and insertions/deletions (InDels), we combined mpileup and call from the bcftools (v1.9) package [64,65]. Here we used the following flags "--redo-BAQ --min-BQ 30 --per-sample-mF" and "--multiallelic-caller --variants-only" respectively. Finally, we only considered variants if they were called significantly in all 3 replicates. We only observed SNPs but no InDels. To confirm these results, we sent the WT Ad5 plasmid (pTG3602) for whole-plasmid sequencing to PlasmidSaurus (Eugene, Oregon).

## Direct RNA sequencing on nanopore arrays

Direct RNA sequencing libraries were generated from 800–900 ng of poly(A) RNA, isolated using the Dynabeads mRNA Purification Kit (Invitrogen, 61006). Isolated poly(A) RNA was subsequently spiked with 0.3 μl of a synthetic Enolase 2 (ENO2) calibration RNA (Oxford Nanopore Technologies Ltd.) and prepared for sequencing using standard protocol steps previously described [58,67]. Sequencing was carried out on a MinION MkIb with R9.4.1 (rev D)

flow cells (Oxford Nanopore Technologies Ltd.) for 20 hours and generated 550,000–770,000 sequence reads per dataset. Raw fast5 datasets were basecalled using Guppy v6.0.1 -c rna_r9.4.1_70bps_hac.cfg -r --calib_detect --trim_strategy rna --reverse_sequence true and subsequently aligned against the adenovirus Ad5 reference genome (AC_000008.1) using MiniMap2.24 (-ax splice -k14 -uf --secondary = no), a splice aware aligner [69]. Resulting SAM files were parsed using SAMtools v1.9 [115].

### Defining TSS and CPAS

Transcription start sites (TSS) as well as RNA cleavage and polyadenylation sites were identified as follows. Sorted BAM files containing sequence reads aligned to the Ad5 genome were parsed to BED12 files using BEDtools [116], separated by strand, truncated to their 5' and 3' termini, and output as BED6 files. Peak regions denoting TSS and CPAS were identified using the HOMER [117] findpeaks module (-o auto -style tss) using a --localSize of 100 and 500 and --size of 15 and 50 for TSS and CPAS, respectively. TSS peaks were compared against Illumina annotated splice sites to identify and remove peak artefacts derived from local alignment errors around splice junctions. Only TSS and CPAS peaks found in both 12 hpi and 24 hpi datasets were used for downstream analysis, except the TSS for the L4 promoter which was only detected at 12 hpi. To predict CPAS sites on the viral genome, we also used the RNA-seq aligner ContextMap2 (version 2.7.9) [70] which has poly(A) read mapping implemented on our short-read data. To run this tool, we used the following optional flags "-aligner_name bowtie–polyA–strandspecific". Due to the previously reported errors when using ContextMap2 at very high read depth, we chose to randomly subsample 10 million, 20 million and 30 million and run the tool on each of the subsets. We only report poly(A) sites if they were called in all three replicates and in at least two of the subsample groups.

### Splice junction correction and sequence read collapsing

Illumina-assisted correction of splice junctions in direct RNA-Seq data was performed using FLAIR v1.3 [72] in a stranded manner. Briefly, Illumina reads aligning to the Ad5 genome were split according to orientation and mapping strand [-f83 & -f163 (forward) and -f99 & -f147 (reverse) ] and used to produce strand-specific junction files that were filtered to remove junctions supported by less than 100 Illumina reads. Direct RNA-Seq reads were similarly aligned to the Ad5 genome and separated according to orientation [-F4095 (forward) and -f16 (reverse) ] prior to correction using the FLAIR correct module (default parameters). Resulting BED12 files were parsed to extend the termini of each individual sequence read to the nearest TSS and CPAS with BlockStarts and BlockSizes (BED12 cols 11 & 12) corrected to reflect this. BED12 files were subsequently collapsed by identifying all reads sharing the same BlockStarts and BlockSizes and reducing these to a single representative. Resulting data were visualized along with the raw read data using IGV [114] and low abundance isoforms (supported by less than 500 junctional reads or 10 full-length reads from Illumina or nanopore data, respectively) removed prior to producing the final annotation.

### Isoform counting

Using our new Ad5 annotation, we generated a transcriptome database by parsing our GFF3 file to a BED12 file using the *gff3ToGenePred* and *genePredtoBED* functions within UCSCutils (https://github.com/itsvenu/UCSC-Utils-Download) and subsequently extracting a fasta sequence for each transcript isoform using the *getfasta* function within BEDtools [116]. Direct RNA-Seq reads were then aligned against the transcriptome database using parameters

optimized for transcriptome-level alignment (minimap2 -ax map-ont -p 0.99). Isoform counts were generated by filtering only for primary alignments (SAM flag 0) with a mapping quality (MapQ) $> 0$.

## Supporting information

**S1 Table. Nucleotide locations of TSS and CPAS.** Nucleotide locations for transcript start sites (TSS) and cleavage and polyadenylation sites (CPAS) within re-annotated transcriptome.
(XLSX)

**S2 Table. Comparison between novel transcripts identified in this study and recent long-read sequencing studies of adenovirus.** Table comparing the presence or absence of the 35 novel transcripts published in this study versus Donovan-Banfield et al. 2020 (Ad5 in MRC5 fibroblasts) and Westergren Jakobsson et al 2021 (Ad2 in IMR90 fibroblasts).
(XLSX)

**S1 Fig. Expanded adenoviral transcriptome.** Expanded transcriptome showing all high confidence transcripts. Color scheme is the same as **Fig 2**. All transcripts are labeled as denoted in **Fig 3**.
(TIF)

**S2 Fig. E4orf6 and E4orf6/DBP have different localization patterns. (A)** The retroviral plasmid pLNE4 was designed to allow the expression of transgenes under the control of the Ad5 E4 promoter. This has the effect of only expressing transgenes after viral infection and E1A expression, thus allowing for the regulated expression of potentially toxic viral proteins. **(B)** A549 cells were transduced with a pLNE4 vector expressing flag-tagged E4orf6. Uninfected cells (open arrowhead) show no expression of flag-tagged proteins. DAPI staining marks nuclei. **(C)** Same as in panel **B**, but A549 cells were transduced with a pLNE4 vector expressing flag-tagged E4orf6/DBP. While E4orf6 is broadly nucleoplasmic, E4orf6/DBP strongly localizes to viral replication centers marked by the cellular protein USP7 (magenta). Immunofluorescence was performed 24 hours post-infection.
(TIF)

**S3 Fig. N-terminal E4orf6 localization of diverse E4 mutant viruses. (A)** A549 cells were infected with the indicated mutant adenoviruses for 24 hours and proteins were resolved by immunoblot. RSA3 antibody shows various banding patterns for E4orf6/DBP, E4orf6, and E4orf6/7. * denotes E4orf6 frameshift product that is produced downstream of the common E4orf6 splice donor in *dl*1002. **(B)** A549 cells were infected with the viruses from **A** for 24 hours before immunofluorescence was performed. Cells were stained with antibodies against E4orf6 N terminal domain (RSA3, green) or cellular USP7 as a marker of viral replication centers (magenta). White scale bar denotes 10 μm.
(TIF)

**S4 Fig. Loss of E4orf6/7 does not result in a small plaque phenotype. (A)** HEK293 cells were infected with limiting dilution of WT Ad5 or ΔE4orf6/7 (*dl*356) for six days to allow the formation of plaques. Plaque formation was negative stained with crystal violet and imaged. Scale bar shows 5 mm. **(B)** WT Ad5 or ΔE4orf6/7 plaque size in HEK293 cells was quantified with ImageJ and normalized to the median plaque size in WT infection. Individual plaques are plotted as a single point. Red error bars denote median and interquartile range. Statistical significance was performed using non-parametric Mann-Whitney t-test (n.s., not significant).
(TIF)

## Acknowledgments

We thank members of the Weitzman, Depledge, and Mohr/Wilson Labs for insightful discussions and input. We are grateful to A. Berk, P. Hearing, G. Ketner, A. Levine, D. Ornelles, and J. Wilson for generous gifts of viruses, antibodies, and reagents. We thank the Upenn Cell and Developmental Biology Microscopy Core for imaging assistance.

## Author Contributions

**Conceptualization:** Alexander M. Price, Matthew D. Weitzman.

**Data curation:** Alexander M. Price, Katharina E. Hayer, Daniel P. Depledge.

**Formal analysis:** Alexander M. Price, Robert T. Steinbock, Richard Lauman, Matthew Charman, Katharina E. Hayer, Daniel P. Depledge.

**Funding acquisition:** Alexander M. Price, Angus C. Wilson, Benjamin A. Garcia, Matthew D. Weitzman.

**Investigation:** Alexander M. Price, Robert T. Steinbock, Richard Lauman, Matthew Charman, Namrata Kumar, Edwin Halko, Krystal K. Lum, Monica Wei, Daniel P. Depledge.

**Methodology:** Alexander M. Price.

**Visualization:** Alexander M. Price.

**Writing – original draft:** Alexander M. Price, Matthew D. Weitzman.

**Writing – review & editing:** Alexander M. Price, Robert T. Steinbock, Richard Lauman, Angus C. Wilson, Daniel P. Depledge, Matthew D. Weitzman.

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
