## [Decision Letter · Decision Letter 0]

19 May 2022

Dear Professor Weitzman,

Thank you very much for submitting your manuscript "Novel viral splicing events and open reading frames revealed by long-read direct RNA sequencing of adenovirus transcripts" for consideration at PLOS Pathogens. As with all papers reviewed by the journal, your manuscript was reviewed by members of the editorial board and by several independent reviewers. In light of the reviews (below this email), we would like to invite the resubmission of a significantly-revised version that takes into account the reviewers' comments.

All three reviewers were enthusiastic about the data presented, and highlighted the novelty in the study. However,  reviewers also agreed that additional experiments should be done, including comparing the phenotypes observed with the E4orf6/DBP mutant virus with an E4orf6/7 mutant virus and correlating the Rsa3/M45 immunofluorescent staining pattern with the presence of the E4orf6-related fusion proteins in order to more strongly support the contention that E4orf6 is not localized to viral replication centers.  Incorporating these additional experiments would strengthen the manuscript and further support conclusions made by the authors.  In addition, it is important to address all of the reviewers concerns in the revision and in the response to reviewers.  

We cannot make any decision about publication until we have seen the revised manuscript and your response to the reviewers' comments. Your revised manuscript is also likely to be sent to reviewers for further evaluation.

Sincerely,

Donna Neumann, PhD

Associate Editor

PLOS Pathogens

Karl Münger

Section Editor

PLOS Pathogens

Kasturi Haldar

Editor-in-Chief

PLOS Pathogens

orcid.org/0000-0001-5065-158X

Michael Malim

Editor-in-Chief

PLOS Pathogens

orcid.org/0000-0002-7699-2064

All three reviewers were enthusiastic about the data presented, and highlighted the novelty in the study. However, the reviewers also agreed that additional experiments should be done, including comparing the phenotypes observed with the E4orf6/DBP mutant virus with an E4orf6/7 mutant virus and correlating the Rsa3/M45 immunofluorescent staining pattern with the presence of the E4orf6-related fusion proteins in order to more strongly support the contention that E4orf6 is not localized to viral replication centers.

Reviewer's Responses to Questions

**Part I - Summary**

Reviewer #1: Manuscript by Price et al is about identification of novel HAdV5 transcripts by using advanced long-read RNA sequencing approach. The study identifies previously missed SNPs in the HAdV5 genome, re-defines TSS & CPAS, and most importantly identifies at least 35 novel transcripts (out of which ca. half of them encode detectable proteins). The manuscript focuses on one of the novel transcript encoded protein, E4orf6/DBP, and elegantly characterizes the functions of this protein in HAdV infection.

The manuscript is very-well written, the authors include correct references and the experiments are well-controlled. I would also highlight that the images are aesthetically pleasing to look at. All of this is not very surprising considering the previous high-impact and high-standard manuscripts coming from the Dr. Weitzman's lab. A big respect for that.

I can not really find major weaknesses of the study, as the experiments are elegantly done and proved using different experimental approaches. Moreover, some of the sequencing results partially overlap with the findings from 2 previous long-read sequencing studies (Donovan-Banfield., 2020 & Westergren-Jakobsson.,2021), hence supporting the reproducibility of the authors sequencing data.

The novelty is clearly in the characterization of one of the new proteins (E4orf6/DBP), something that previous sequencing studies did not even consider. Or in other words, we all have missed this protein, although it was always on our western blot images with an anti-E4orf6 antibody! Moreover the authors also make a fundamental finding regarding the E4orf6 subcellular localization, which the adenovirus community has been fooled for decades due to detection of the E4orf6/DBP protein. Additional mapping of the TSS and CPAS will help the adenovirus community to better characterize the synthesis and co-/post-transcriptional processing of the viral transcripts, something that is clearly needed to get a better picture of the HAdV life-cycle. And finally, identification of novel viral transcripts/proteins, which will take now some time to find their function.

Summa summarum: as such the manuscript will not have only a very high impact among the adenovirologists but also among other DNA virus researchers studying the complexity of viral transcriptomes.

Reviewer #2: The authors utilize Illumina short-read and nanopore long-read direct RNA sequencing approaches in Adenovirus type 5 (HAdV-C5)-infected A549 cells to identify all viral transcripts including transcription start sites, RNA cleavage, and RNA polyadenylation sites. Several published reports (refs 54 and 55) previously mapped HAdV-C2/5 transcripts during time courses of infection, but the current study identified a number of new transcripts not previously reported. Here, the authors identified 35 new HAdV-C5 viral transcripts including 14 with new splice junctions, 6 novel ORF mRNAs, and 15 mRNAs with splicing patterns that could generate new fusion proteins of annotated ORFs. The authors pursue one such novel fusion ORF that fuses the N-terminal region of E4orf6 with the E2-encoded DNA binding protein (DBP) - termed E4orf6/DBP. They confirm the expression of E4orf6/DBP in infected cells and generate a mutant virus that eliminates its expression. The E4orf6/DBP mutant virus replicates normally and expresses viral proteins equivalently to HAdV-C5 wild-type virus. The authors identify two phenotypes associated with loss of E4orf6/DBP expression: viral nuclear replication centers have an unusual morphology and the virus exhibits a small plaque phenotype. Overall, these studies are convincing and technically at the cutting edge. With this said, the results are highly specific to Adenovirus and thus directed toward a limited readership. Also, the basis of the altered replication center morphology and small plaque phenotype with the E4orf6/DBP viral mutant is unknown.

Reviewer #3: Long-read (nanopore) and short-read (Illumina) sequencing reveals three dozen novel viral transcripts from human adenovirus type 5. Of note, a novel evolutionarily conserved protein containing the amino-terminus of E4orf6 fused to the E2A DNA-binding protein was identified. Expression of this conserved fusion protein was eliminated by changing the E4orf6/7 splice-donor site (also eliminating the non-essential E4orf6/7 fusion protein.) The absence of this novel fusion protein was found to have no measurable consequences on viral gene expression or genome replication but may have affected the manner or rate of cell death or virus release.

This work addresses a critical gap in our knowledge. Specifically, the annotation of the type 5 reference strain is incomplete and lacks transcription start sites, RNA cleavage and polyadenylation sites and the associated 5’ and 3’ non-translated sequences. This work also provides new information on novel adenovirus products that encourage a re-evaluation of our previous understanding of the localization and distribution of key viral proteins during a viral infection.

Recently published work from Mathews (2020) and Akusjarvi (2021) have described the complexity of transcription from adenovirus type 2 and type 5 using long-read methods. This work provides similar information on the transcriptional profile of type 5 adenovirus but includes a comprehensive proteomic analysis and integrated short-read analysis to define transcription start sites as well as polyadenylation and cleavage sites. An unappreciated fusion product was verified in this study at the level of RNA sequencing and protein analysis. The contribution (or lack thereof) of this fusion protein to viral replication was studied. Although this E4orf6/E2A-DPB fusion product appears to have no effect on viral replication, it does appear to contribute to a larger plaque phenotype, consistent with either promoting cell death or virus release.

The quality and the breadth of the data presented here as well as the comprehensive and cogent review of the pertinent literature is a significant strength of this report. Another strength of this work that differentiates this from previously published catalogs of transcriptional changes during adenovirus infection is the coordinated analysis of RNA and protein. The singular limitation I can identify is the overly enthusiastic interpretation that the novel findings here undermine previous held views on adenovirus protein localization with implications for the function of those proteins.

**Part II – Major Issues: Key Experiments Required for Acceptance**

Reviewer #1: I can not really find any major issues with the paper. Experimentally it is very well conducted study with proper controls and repetitions.

Reviewer #2: The E4orf6/DBP mutant virus also does not express the E4orf6/7 protein. The authors mention that an E4orf6/7 mutant virus replicates normally and does not have any obvious phenotype, but does this include an examination of viral replication center morphology and/or plaque size under conditions used in these studies? It seems entirely possible that the phenotypes observed with the E4orf6/DBP mutant virus may partly be due to the loss of E4orf6/7. An E4orf6/7 mutant virus should be examined for comparison.

Reviewer #3: It is possible that staining at replication centers is dominated by the E4orf6/E2A-DBP fusion protein. If this is so, the viral replication centers in cells infected with E4orf6-deletion virus (dl355) studied by immunoblot in Fig. 6 should show prominent Rsa3/M45 staining patterns at the viral replication centers similar to that seen in cells infected with the wild-type virus. If this is so, this would strongly support the notion that the bulk of that staining pattern is due to the novel fusion protein. I suggest that this experiment be performed along with careful attention to correlating the Rsa3/M45 immunofluorescent staining pattern with the presence of the E4orf6-related fusion proteins in order to more strongly support the contention that E4orf6 is not localized to viral replication centers. The statement that previous reports suggesting localization of E4orf6 to the viral replication was incorrect may be a bit overstated without additional evidence.

The statement in lines 376-377, “Now that we know a major fraction of the E4orf6 signal detected by Rsa3 or M45 during a viral infection is actually E4orf6/DBP…” is somewhat misleading when used to re-interpret previous published results on the localization of E4orf6. Specifically, this report (Fig. 4E) shows that that this novel fusion form is not present at 16 h postinfection and appears to be lower in abundance compared to the E4orf6 form at 24 h postinfection. The implication that “a major fraction of the E4orf6 signal” is due to the fusion protein needs to be qualified accordingly by noting that this can be true only at significantly late times of infection. Previously conclusions from reports describing the localization of E4orf6 at or before 16 h postinfection are not negated by this report.

**Part III – Minor Issues: Editorial and Data Presentation Modifications**

Reviewer #1: I have the following minor comments:

L218-219: I strongly suggest to make Supplementary table where the authors will point out the exact nucleotides they have identified as the TSS and CPAS. Something that was done in the paper by Donovan-Banfield., 2020 (the table is in the supplementary part in their paper). The present figure 2 is nice to look at but does not tell the exact sites/nucleotides as it would be important to compare it with the study by Donovan-Banfield., 2020. Further, the exact TSS & CPAS will help to study the details of HAdV RNA transcription and processing.

Extra: In the paper by Donovan-Banfiled., the authors indicated that TSS identified by the Nanopore sequencing was ~8–15 nucleotides downstream of the previously established locations. That was regarded as a Nanopore sequencing specific phenomena/artefact. So, it would be very informative to know if the Price et al., also saw the similar phenomena? Is that taken into consideration when you were defining the TSS? Again, the supplementary table with exact nt. for TSS and CPAS would help a lot here.

L300: to be honest, I can not see the "increase dramatically" effect for the E4-DPB and E4orf6/DBP due to low resolution of the graphs. I would suggest to include numbers for fold increase (12hpi vs 24hpi) after each transcript (smth like: 19K/IX(4x), E4-DPB(Xx), E4orf6/DBP(Xx)) to get a better idea about the increase. And avoid word "dramatically"....

L401: I suggest to point out that the loss of E4orf6 had a major effect at 24hpi, and partial effect at 48hpi based on the provided image. To honest, I do not see major effect of E4orf6 deletion on hexon and fiber at 48hpi.

L403: it is overstatement, consider using "partial loss of degradation" as based on the provided image Rad50 level is still less in E4orf6 lacking cells when compared to mock cells.

L487: not entirely true considering 2 previous studies by Donovan-Banfield., 2020 & Westergren-Jakobsson.,2021. Consider rephrasing the sentence.

L502-510: The study Westergren-Jakobsson.,2021 also identified SNPs in the HAdV2 with Nanopore sequencing. The authors should compare, and discuss, the SNPs found in their study to the ones found by Westergren-Jakobsson.,2021. Are there any overlaps?

Discussion in general: the authors are modest in describing the previous studies by Donovan-Banfield., 2020 & Westergren-Jakobsson.,2021. (L516). It would be more informative if the authors can discuss if any of their novel 35 transcripts were also identified in these 2 previous studies. Or at least comment if the E4orf6/DBP transcript was detected in HAdV2/IMR90 (Westergren-Jakobsson.,2021.) and HAdV5/MRC-5 (Donovan-Banfield., 2020) virus/cell systems. That may give a hint about the universal nature of the E4orf6/DBP transcript in different cell systems.

Reviewer #2: The authors conclude on line 386 that E4orf6/DBP has "different cellular localizations, binding partners, and functions compared to E4orf6." They clearly show overt differences in subcellular localization, but on what basis do they conclude that E4orf6/DBP has different binding partners and functions? They can discriminate binding of E4orf6 with E1B55K protein and Mre11/Rad50 degradation, but this does not mean E4orf6/DBP has novel binding partners and function.

Reviewer #3: Localization of the FLAG-tagged E4orf6 variants is another concern. The results presented in Suppl. Fig 2 were argued to support the notion that E4orf6 is diffusely localized in the nucleus and does not strongly association with viral replication centers. How long after infection were these result analyzed and how well does the retroviral construct recapitulate the time course of expression of the various E4 and E2A-DBP forms?

Another concern is that anecdotal evidence has suggested that N-terminal modifications of E4orf6 create a non-functional protein. Is the FLAG-tagged E4orf6 protein studied here known to be functional with respect to viral replication (and not simply reconstituting the ubiquitin protein ligase?)

How does the failure of E1B-55K to localize to viral replication centers in the absence of E4orf6 reconcile with the absence of E4orf6 at the viral replication centers?

Any speculation on why internal L4 promoter identified by Leppard not identified in A549 cells?

PLOS authors have the option to publish the peer review history of their article (what does this mean?). If published, this will include your full peer review and any attached files.

Reviewer #1: No

Reviewer #2: No

Reviewer #3: No
---

## [Editor Report · Decision Letter 1]

5 Aug 2022

Dear Matt 

We are pleased to inform you that your manuscript 'Novel viral splicing events and open reading frames revealed by long-read direct RNA sequencing of adenovirus transcripts' has been provisionally accepted for publication in PLOS Pathogens.

Best regards,

Donna Neumann, PhD

Associate Editor

PLOS Pathogens

Karl Münger

Section Editor

PLOS Pathogens

Kasturi Haldar

Editor-in-Chief

PLOS Pathogens

orcid.org/0000-0001-5065-158X

Michael Malim

Editor-in-Chief

PLOS Pathogens

orcid.org/0000-0002-7699-2064
---

## [Editor Report · Acceptance letter]

8 Sep 2022

Dear Dr. Weitzman,

We are delighted to inform you that your manuscript, "Novel viral splicing events and open reading frames revealed by long-read direct RNA sequencing of adenovirus transcripts," has been formally accepted for publication in PLOS Pathogens.

Best regards,

Kasturi Haldar

Editor-in-Chief

PLOS Pathogens

orcid.org/0000-0001-5065-158X

Michael Malim

Editor-in-Chief

PLOS Pathogens

orcid.org/0000-0002-7699-2064